# DR3 stimulation of adipose resident ILC2s ameliorates type 2 diabetes mellitus

Pedram Shafiei-Jahani[1], Benjamin P. Hurrell[1], Lauriane Galle-Treger[1], Doumet Georges Helou [1], Emily Howard[1], Jacob Painter [1], Richard Lo[1], Gavin Lewis[2], Pejman Soroosh[2] & Omid Akbari [1]✉

Disturbances in glucose homeostasis and low-grade chronic inflammation culminate into metabolic syndrome that increase the risk for the development of type 2 diabetes mellitus (T2DM). The recently discovered group 2 innate lymphoid cells (ILC2s) are capable of secreting copious amounts of type 2 cytokines to modulate metabolic homeostasis in adipose tissue. In this study, we have established that expression of Death Receptor 3 (DR3), a member of the TNF superfamily, on visceral adipose tissue (VAT)-derived murine and peripheral blood human ILC2s is inducible by IL-33. We demonstrate that DR3 engages the canonical and/or non-canonical NF-κB pathways, and thus stimulates naïve and co-stimulates IL-33-activated ILC2s. Importantly, DR3 engagement on ILC2s significantly ameliorates glucose tolerance, protects against insulin-resistance onset and remarkably reverses already established insulin-resistance. Taken together, these results convey the potent role of DR3 as an ILC2 regulator and introduce DR3 agonistic treatment as a novel therapeutic avenue for treating T2DM.

───────────
[1] Department of Molecular Microbiology and Immunology, Keck School of Medicine, University of Southern California, Los Angeles, CA, USA. [2] Janssen Research and Development, San Diego, CA, USA. ✉email: akbari@usc.edu

According to the World Health Organization (WHO), the rate of obesity has approximately tripled in the past four decades[1]. Excess adiposity has been implicated as the driving factor for the development of various metabolic disorders such as hypertension, nonalcoholic fatty liver disease, insulin resistance and type 2 diabetes mellitus (T2DM). Numerous studies have clearly demonstrated that leukocytes induce obesity-associated low-grade chronic inflammation, which leads to insulin resistance, elevation of blood glucose levels and development of T2DM[2]. This inflammation and metabolic disturbance result in expansion and remodeling of the leukocytes in tissues that regulate energy homeostasis, such as visceral adipose tissue (VAT)[3]. Recently, group 2 innate lymphoid cells (ILC2s) have been described as one of the principle immune cells present in VAT[4,5]. Due to their ability to secrete copious amounts of IL-5 and IL-13, ILC2s has become the center of attention in the scientific community for regulating metabolic homeostasis in VAT[6]. ILC2-derived IL-13 leads to the differentiation of macrophages toward an anti-inflammatory phenotype, referred to as alternatively activated macrophages (AAMs). Moreover, ILC2-derived IL-5 leads to activation and recruitment of eosinophils, which in turn secrete large quantities of IL-4 that is required for maintenance of AAMs[4]. AAMs, also known as "M2" macrophages, are a subset of anti-inflammatory macrophages that are defined by the production of IL-10, arginase 1, CD206 (mannose receptor), and are important contributors to adipose tissue remodeling and modulation of inflammation[7]. Insufficient ILC2 activity has been associated with decreased numbers of AAM in the adipose tissue, which has been shown to promote insulin resistance[4,6].

Death Receptor 3 (DR3), also known as tumor necrosis factor receptor superfamily member 25 (TNFRSF25) is a member of the TNFR superfamily present on the cell surface of various leukocytes, including T cells, mucosal ILCs, NK cells, and myelocytes such as macrophages[8–15]. In addition, various expression levels of DR3 are observed on nonimmune related cell populations such as neurons, stromal cell, such as fibroblasts, and epithelial cells[16–19]. DR3 is a type 1 membrane protein that contains a death domain in its cytoplasmic region. Depending on the cell type, DR3 has been reported to participate in different biological processes such as apoptosis or co-stimulation of T cell responses[20,21]. Antigen presenting cells (APC)-derived TL1A (Tumor necrosis factor-like cytokine 1A), a type II transmembrane protein, has been previously described as the only TNF-superfamily ligand for DR3 in vivo and is present in inflamed tissues[21–23]. Furthermore, Progranulin-derived Atsttrin has also been shown to bind DR3 and inhibit the TL1A/DR3 axis[24]. Previous human genetics and mouse studies have implicated the TL1A/DR3 axis in various chronic immunological disorders, such as Inflammatory Bowel Disease (IBD), Rheumatoid Arthritis, and allergic asthma[22,23,25]. Our group and others have previously demonstrated that activation of VAT ILC2s can limit adiposity and insulin resistance in mice fed a high fat diet (HFD)[4,6,26]. The specific role of DR3 signaling in T2DM and metabolic homeostasis has not been investigated.

In this study, we have evaluated the mechanism, signaling, and therapeutic potentials of DR3-depedent stimulation of VAT-derived ILC2s in the context of insulin-resistance and T2DM. We have found both peripheral blood human and VAT-derived mouse ILC2s express DR3, and this expression is further increased by IL-33. Interestingly, DR3 engagement induces type 2 cytokine production in both naïve and activated ILC2s, suggesting both a stimulatory and co-stimulatory role for DR3. We demonstrate here that DR3 engagement induces both the canonical and non-canonical NF-κB pathways in naïve ILC2s. However, DR3 engagement on IL-33-activated ILC2s only induces the non-canonical NF-κB pathway as evidenced by upregulation of kinase (NIK, encoded by *Map3k14*), *Nfkb2* (p52) and *Relb*.

Furthermore, utilizing a DR3-specific agonist antibody, we establish both a preventative and therapeutic role for DR3 engagement on ILC2s in the context of T2DM. Importantly, we demonstrate that the potent therapeutic effects of DR3 engagement are IL-13 dependent, as demonstrated by adoptive-transfer of IL-13 deficient ILC2s. Our results further emphasize the critical role of DR3 as a modulator of ILC2 effector functions, thus providing new insights into the role of DR3 in VAT-derived ILC2s and establishing DR3 agonist treatment as a promising and novel therapeutic option for prevention and treatment of T2DM.

## Results

**DR3 engagement on ILC2s promotes type 2 cytokine secretion.** Since ILC2s have been recently recognized for their central role in both initiating and perpetuating inflammation, there have been growing interests and efforts to identify ILC2 biomarkers that can modulate ILC2 activity and serve as targets in the future therapeutics. Our group recently reported that TNFR2[27] and GITR[28,29], two members of the TNFR superfamily, are expressed by ILC2s and can modulate their effector functions and regulate homeostasis. In this vein, we first investigated whether DR3 was similarly expressed on naïve and activated VAT ILC2s. A cohort of C57BL/6J mice were fed a normal chow diet (NCD) or a HFD to induce obesity for 14 weeks. Subsequently, the mice received recombinant IL-33 to activate ILC2s or PBS intraperitoneally for three consecutive days (Fig. 1a). On the fourth day, the visceral adipose tissue was isolated and ILC2s were analyzed by fluorescence-activated cell sorting (FACS) and gated as lineage $^-$CD45$^+$IL-7R$^+$ and ST2$^+$ (Fig. 1b). FACS analysis of the cell-surface phenotype of VAT-derived ILC2s revealed that DR3 is expressed on naïve ILC2s regardless of the diet, and this expression is further inducible by IL-33 (Fig. 1c). To investigate the effect of DR3 stimulation on ILC2s, we subsequently measured the cytokine secretion levels of ILC2s in the presence of DR3 agonist or isotype control antibody. Freshly isolated VAT-derived naïve and in vivo IL-33-activated ILC2s were cultured ex vivo for 48 h with either DR3 agonist or corresponding isotype controls. The cytokine secretion levels were then measured in the cell culture supernatants by Luminex assay. Interestingly, cytokine levels of IL-5, IL-13, GM-CSF, and MIP2 were augmented by DR3 engagement on naïve ILC2s compared to controls (Fig. 1d). However, DR3 treatment did not induce IL-6 and IL-9 in naïve ILC2s. Importantly, DR3 engagement on in vivo activated ILC2s strikingly enhanced the cytokine secretion levels of the aforementioned cytokines, including IL-6 and IL-9 (Fig. 1d). Taken together, these results suggest that DR3 is expressed on VAT-derived naïve and IL-33-activated ILC2s and acts as a stimulatory or co-stimulatory molecule, respectively.

**DR3 engagement protects against onset of type 2 diabetes.** Our group and others have previously demonstrated that activation of VAT ILC2s can limit adiposity and insulin resistance in mice fed a HFD[4,6,28]. Thus, we next explored whether DR3-dependent activation of ILC2s could similarly prevent the development of insulin-resistance in vivo. A cohort of C57BL/6J mice were fed either a normal chow diet (NCD) or an HFD to induce obesity. Furthermore, the mice were intraperitoneally treated with either DR3 agonist (1 mg/mouse) or the isotype control every four days for 14 weeks and various metabolic parameters were assessed (Fig. 2a). We observed a similar increase in weight gain in both the DR3 agonist-treated and isotype control-treated cohorts that were placed on the HFD and both developed diet-induced obesity when compared to the NCD fed mice (Fig. 2b). However, the DR3 agonist-treated mice had significantly lower fasting blood glucose levels compared to isotype-treated mice placed on a HFD

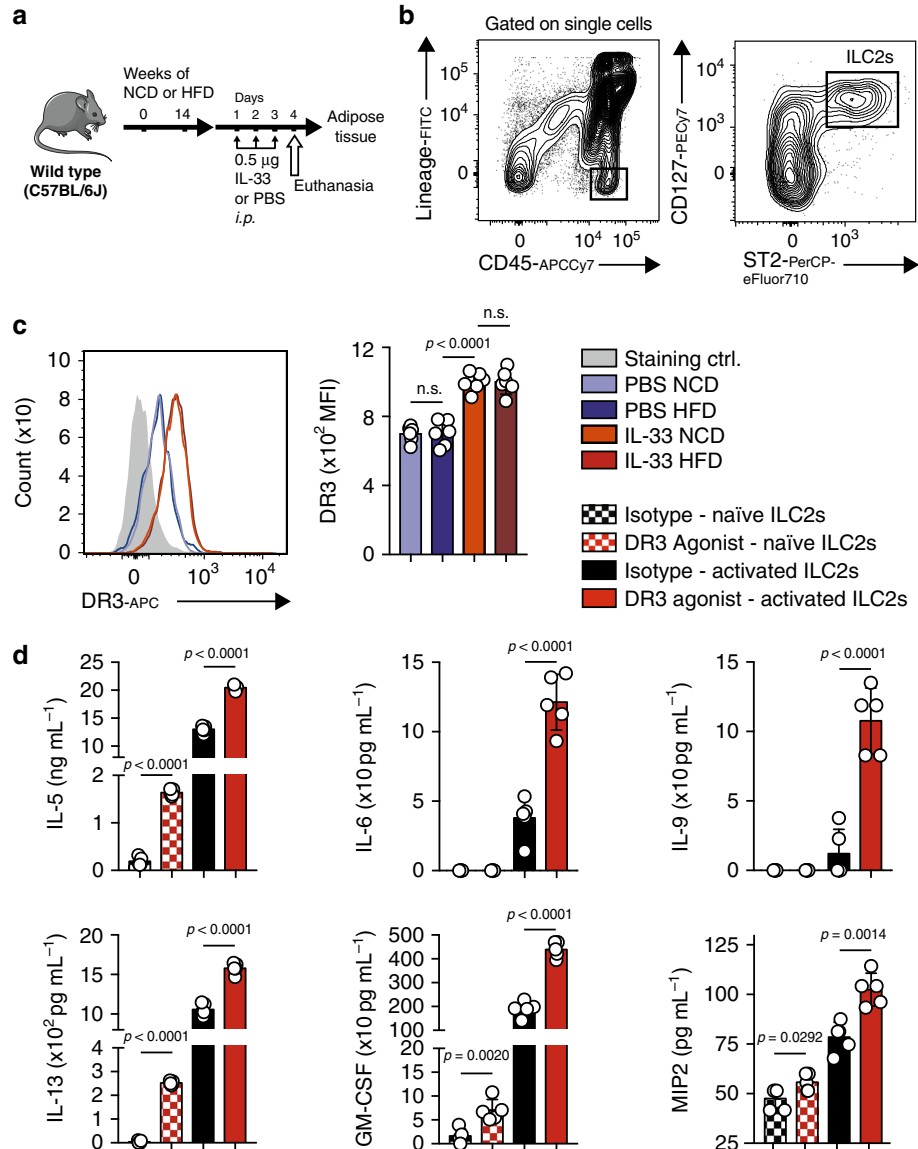

**Fig. 1 DR3 agonistic treatment induces type 2 cytokines in both naïve and activated ILC2s. a** A cohort of C57BL/6J mice were fed a normal chow diet (NCD) or a high fat diet (HFD) for 14 weeks and were subsequently challenged with recombinant mouse (rm) IL-33 (0.5 μg in 50 μL) or PBS intraperitoneally (i.p.) for 3 days, $n = 6$ mice. The mice were euthanized on fourth day and the visceral adipose tissue (VAT) was isolated, as shown in the timeline. **b** Gating strategy of Lin[-]CD45[+]IL-7R[+]ST2[+] ILC2 cells. **c** DR3 expression in naïve and IL-33-activated murine white adipose tissue-derived ILC2s compared to the isotype control. Quantitation of DR3 expression is shown as MFI ± SD. **d** Freshly sorted naïve and activated VAT ILC2s were cultured in the presence of recombinant mouse (rm) IL-2 and rmIL-7 stimulated with DR3 agonist (5 μg/mL) or isotype control for 48 h. The levels of IL-5, Il-6, IL-9, IL-13, GM-CSF, and MIP2 were measured by Luminex on the culture supernatants, $n = 5$ mice. Error bars are the mean ± SD. Statistical analysis, one-way ANOVA (**c**), two-tailed student's $t$-test (**d**); n.s.: $p$-value of <0.05 was considered as non-significant. Mouse image provided with permission from Servier Medical Art.

(Fig. 2c). As indicated by the intraperitoneal glucose tolerance tests (ip-GTTs) and insulin tolerance tests (ITTs), the glucose tolerance (Fig. 2d) and insulin sensitivity (Fig. 2e) of the DR3 agonist-treated cohort significantly improved when compared to the isotype-treated control. Analysis of the same metabolic parameters in BALB/c mice (Supplementary Fig. 1) are in agreement with the findings in C57BL/6J mice, indicating DR3-depedent stimulation of ILC2s significantly limits the onset of obesity, improves glucose homeostasis and insulin resistance in mice independent of the genetic background. Additionally, we explored the source of IL-33 that activates ILC2 in vivo during chronic low-grade inflammation caused by diet-induced obesity in VAT lysates by quantitative real time PCR. Compared to

NCD-treated mice that expressed low levels of IL-33 transcripts over the 14-week treatment, we observed a progressive significant increase in mRNA levels of IL-33 in HFD-treated mice. Since cellular damage-associated signals can regulate IL-33 protein secretion, these transcriptomic results are suggestive of potential VAT IL-33 secretion in the local environment (Fig. 2f). We next assessed the presence of ILC2s in VAT throughout the DR3 regimen. Both the number of ILC2s (Fig. 2g), as well as percentage of ILC2s (Supplementary Fig. 2a) was increased in VAT of HFD fed mice over time. Taken together, these results establish that DR3 stimulation significantly limits the onset of obesity and improves glucose homeostasis in the context of metabolic dysfunctions.

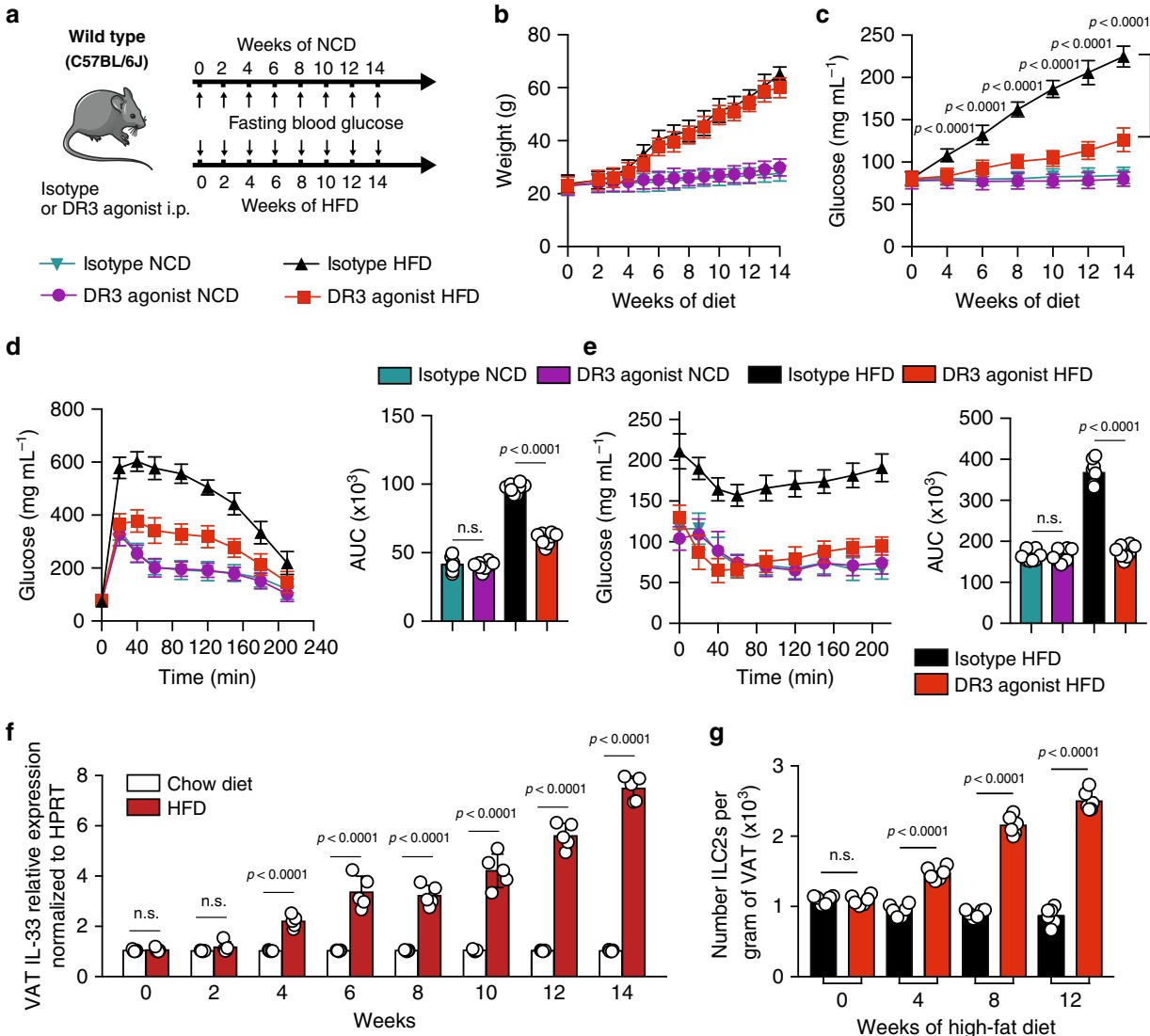

**Fig. 2 DR3 stimulation allows for protection from the onset of type 2 diabetes in WT mice. a** As shown in the timeline, a cohort of C57BL/6J mice were fed a normal chow diet (NCD) or a high fat diet (HFD) and treated with either DR3 agonist (1 mg/mouse) or isotype control via intraperitoneal injections (i. p.) every four days, $n = 8$ mice. **b** The total weight of the mice was measured once every week, and **c** the fasting blood glucose levels were measured once every two weeks for a period of 14 weeks. **d** Glucose tolerance test, and **e** insulin tolerance test were performed on the 14th week on the treatment. The corresponding area under the curve (AUC) was calculated for each cohort. **f** Time-kinetic qPCR expression of IL-33 in VAT lysate. **g** The number of VAT ILC2s in HFD-fed mice throughout the treatment course with DR3 agonist or isotype. Error bars are the mean ± SD. Statistical analysis, one-way ANOVA (**b**, **c**), two-way ANOVA (**d**, **e**), two-tailed student's $t$-test (f and g); n.s.: $p$-value of <0.05 was considered as non-significant. Mouse image provided with permission from Servier Medical Art.

**DR3 engagement in the absence of adaptive immunity prevents T2DM.** Since previous reports have demonstrated that DR3 is expressed by other leukocytes such as helper and cytotoxic T lymphocytes, we next assessed whether the prevention of obesity-induced metabolic disturbances via DR3 stimulation is ILC2-dependent. Thus, we examined the effects of DR3 agonistic treatment in immunodeficient $Rag2^{-/-}$ mice that lack any mature B and T cells. A cohort of $Rag2^{-/-}$ mice was fed a HFD and intraperitoneally treated with either DR3 agonist (1 mg/mouse) or the isotype control every four days for a period of 14 weeks (Fig. 3a). In line with our previous results, DR3 stimulation did not alter weight gain when compared to the isotype control-treated cohorts fed the same diet (Fig. 3b). Similarly, DR3 engagement did not alter the weight of the liver after 14 weeks (Supplement Fig. 2d). However, DR3 treatment strikingly decreased fasting blood glucose concentrations as early as 2 weeks

after onset of treatment (Fig. 3c), as well as insulin concentrations in plasma (Fig. 3d). Moreover, the ip-GTTs and ITTs results indicate that DR3 stimulation significantly increased both glucose tolerance (Fig. 3e) and insulin sensitivity (Fig. 3f) in $Rag2^{-/-}$ mice when compared to isotype-treated cohort. Our group and others have previously established that ILC2 activation can induce beiging of the adipose tissue, which in turn increases thermogenesis and caloric expenditure[4,6,28]. To assess the beiging of the adipose tissue via DR3-dependent stimulation of ILC2s, we next examined the expression levels of uncoupling protein 1 (Ucp1), a critical protein implicated in the thermogenic process of adipocytes. Our results indicate that mice treated with DR3 agonist exhibited significantly enhanced expression of Ucp1 at the transcriptomic level, as evidenced by the RT-qPCR in the VAT (Fig. 3g). Furthermore, our group and others have demonstrated that IL-5 and IL-13 have a protective effect on the development of

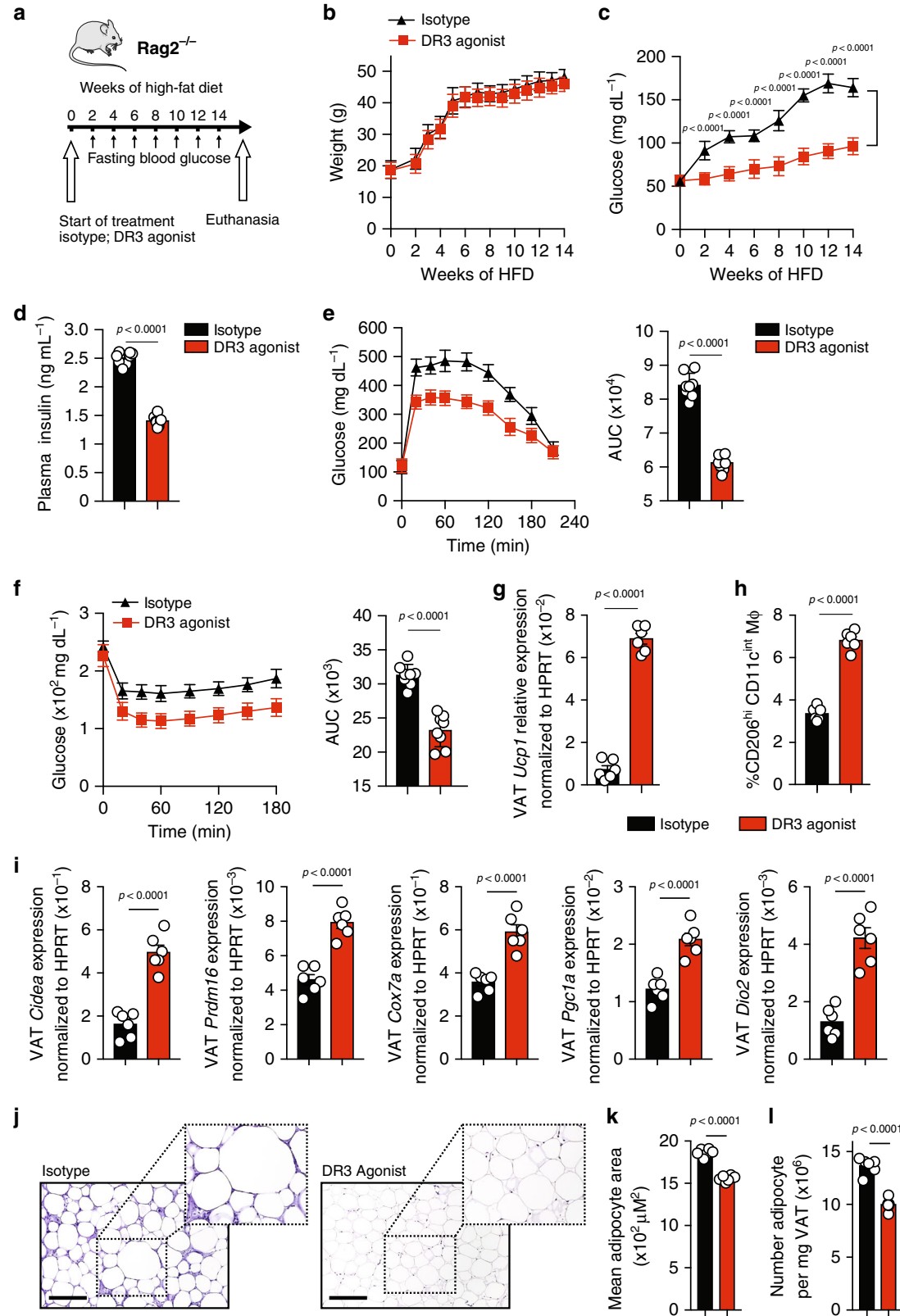

insulin-resistance[4,6,28]. IL-13 directly promotes an alternatively activated macrophage (AAM) phenotype. IL-5 is required for eosinophil recruitment and activation, which further contributes to maintenance of AAMs[26]. As our previous ex vivo results indicate that DR3 engagement promotes IL-5 and IL-13 secretions in naïve and activated ILC2s (Fig. 1d), we next determined if DR3 treatment altered the presence of AAMs in VAT. Our results confirm that DR3 engagement on ILC2s results in an increased presence of VAT-associated AAMs (Fig. 3h). Similarly, the abundance of VAT ILC2s was augmented by DR3 agonistic treatment (Supplementary Figure 2b-c). Moreover, other known genetic markers of adipose beiging were induced by DR3 agonist

**Fig. 3 DR3 engagement allows for preemptive protection from type 2 diabetes in *Rag2*−/− mice. a** A cohort of *Rag2*−/− mice were placed on HFD and intraperitoneally treated with either DR3 agonist (1 mg/mouse) or isotype control every four days, n = 8 mice. **b** The total weight of the mice in each cohort was measured every week, and **c** blood glucose levels were measured every two weeks for a period of 14 weeks. **d** The plasma insulin concentrations were measured using ELISA. **e** Glucose tolerance test and **f** insulin tolerance test were performed at the end of 14 weeks. The corresponding area under the curve (AUC) was calculated for each cohort. **g** *Ucp1* transcripts levels in VAT lysates, and **h** relative amount of CD45+CD11b^hiF4/80^hiCD206+CD11c+ AAM macrophages in VAT after 14 weeks of treatment, n = 6. **i** The mRNA expression of VAT lysates after 14 weeks was determined by qRT-PCR using specific primers for *Cidea*, *Prdm16*, *Pgc1a*, *Cox7a*, *Dio2* and hypoxanthine-guanine phosphoribosyltransferase (*Hprt*). **j** Hematoxylin and eosin-stained epididymal adipose tissue sections (×400, scale bars, 100 μm). Quantitation of the adipocyte area **k** and number **l** observed for each cohort. Error bars are the representative of mean ± SD. Statistical analysis, one-way ANOVA (**b**, **c**), two-way ANOVA (**e**, **f**), two-tailed student's *t*-test (**d**, **g**–**i**, **k**–**l**). Mouse image provided with permission from Servier Medical Art.

treatment in VAT lysate (Fig. 3i). Particularly, higher expression of cell death activator CIDE-A (*Cidea*), PR domain zinc finger protein 16 (*Prdm16*), peroxisome proliferator-activated receptor-γ coactivator (*Pgc1a*), cytochrome oxidase subunit VIIa polypeptide (*Cox7a*), and deiodinase 2 (*Dio2*) was detected in VAT lysates of mice treated with DR3 agonist compared to isotype control cohort. To further explore the effects of DR3 treatment, we next performed histological analysis of VAT structure (Fig. 3j). Both adipocyte hypertrophy and hyperplasia were significantly decreased in the DR3 agonist recipients (Fig. 3k-l). Since *Dio2*, *Prdm16* and *Pgc1a* expressions are also positively associated with a higher metabolic rate[30–32], we next examined 5 well-known genes encoding respiratory chain complexes (I, III, IV, and V) and tyrosine hydroxylase (*Th*) in the adipose tissue of the mice treated with DR3 agonist or isotype. DR3 agonist significantly increased the expression levels of genes including complex I (*Nd1* and *Nd2*), complex III (mitochondria-encoded NADH dehydrogenase I (*Cytb*)), complex IV (mitochondria-encoded cytochrome c oxidase I (*Cox1a*)), complex V (mitochondria-encoded ATP synthase 6 (*Atp6*)) and tyrosine hydroxylase (*Th*) in the adipose tissue (Supplementary Fig. 2e-j). Taken together, these results demonstrate that DR3 stimulation of ILC2s improves glucose homeostasis and protects from onset of type 2 diabetes through beiging of the adipose tissue, abrogating both adipocyte hypertrophy and hyperplasia, and promoting VAT-associated AAM phenotype in the absence of B and T cells in vivo.

**Therapeutic DR3 treatment reverses insulin-resistance**. Given that DR3 stimulation can protect from the onset of type 2 diabetes, we next examined the therapeutic potential of DR3 agonistic treatment in a model of *Rag2*−/− mice with previously established type 2 diabetes. A cohort of *Rag2*−/− mice was placed on HFD for 14 weeks. After insulin-resistance was established at 8 weeks, the mice began treatments with either DR3 agonist (1 mg/mouse) or the isotype control intraperitoneally once every 4 days (Fig. 4a). In agreement with our previous findings, DR3 agonist had no effect on the amount of weight gained over the course of HFD when compared to the isotype control (Fig. 4b). Strikingly, agonistic DR3 treatment substantially reduced both fasting blood glucose concentrations (Fig. 4c) and plasma insulin concentrations compared to isotype control concentrations (Fig. 4d). Furthermore, glucose tolerance (Fig. 4e) and insulin sensitivity (Fig. 4f) were both drastically improved as indicated by the ip-GTTs and ITTs results, respectively. Histological analysis of VAT isolated from DR3 treated group displayed significant improvements in adipocyte hypertrophy and hyperplasia (Fig. 4-i). Collectively, these results suggest anti-DR3 agonistic treatment reversed insulin-resistance in context of established metabolic syndrome.

**Protective effect of DR3 agonist is dependent on ILC2 effector function**. Next, we asked whether DR3 stimulation of ILC2s and

their effector function is sufficient to prevent insulin resistance and regulate glucose homeostasis. ILC2s were isolated from either Wild-Type (WT), *Il5*−/− or *Il13*−/− mice and adoptively transferred into a cohort of *Rag2*−/− *Il2rg*−/− mice that lacked any mature T cells, B cells, NK cells and ILCs. Subsequently, the recipient mice and control cohorts without transfer of cells were placed on an HFD for 14 weeks and were intraperitoneally treated with either DR3 agonist (1 mg/mouse) or the isotype control every four days (Fig. 5a). As expected, the amount of weight gained over the course of HFD was not affected by the DR3 agonistic treatment (Fig. 5b) in any of the cohorts. However, we observed considerably reduced fasting blood glucose concentrations in the cohort treated with DR3 agonist and adoptively transferred with WT ILC2s (Fig. 5c). Moreover, WT ILC2s in these recipients also allowed for prominently enhanced glucose tolerance (Fig. 5d) after treatment with DR3 agonist when compared to the group with adoptively transferred WT ILC2 that were treated with the isotype control. These results further confirm that the protective effect of DR3 stimulation is specifically dependent on the expression of DR3 on ILC2s. Strikingly, the protective effect of DR3 agonistic treatment on glucose tolerance and fasting blood glucose levels completely disappeared in the cohort that were adoptively transferred with IL-13-deficient ILC2s when compared to the WT ILC2s (Fig. 5c, d). Moreover, the protective effect of the DR3 agonistic treatment on fasting blood glucose levels was partially repressed in the group transferred with IL-5 deficient ILC2s (Fig. 5c). DR3 treatment did reduce the VAT weight in cohorts treated with DR3 agonist and adoptively transferred with WT or IL-5 deficient ILC2s when compared to cohorts that were treated with the isotype control or cohorts that were not adoptively transferred with any ILC2s (Fig. 5e). Remarkably, this reduction in VAT weight disappeared in the cohort that was treated with the DR3 agonist but was the recipient of IL-13 deficient ILC2s (Fig. 5e). Lastly, DR3 agonistic treatment significantly increased the number of WT ILC2s per gram of VAT compared to the isotype treatment (Fig. 5f). Intriguingly, the number of ILC2s after DR3 agonistic treatment was significantly abrogated in the recipients of IL-5 deficient and IL-13 deficient ILC2s compared to the WT ILC2 recipient mice (Fig. 5f). Taken together, these data demonstrate that the protective effect of DR3 stimulation of ILC2s is dependent on ILC2s effector function and ILC2-derived secretion of type 2 cytokines such as IL-13.

**DR3 induces canonical/non-canonical NF-κB pathways in ILC2s**. To further explore the molecular mechanisms associated with protective effects of DR3-depedent stimulation of ILC2s, we isolated activated VAT ILC2s from WT mice and cultured them in presence of DR3 agonist or isotype control (5 μg/mL) for 24 h ex vivo. Subsequently, we performed RNA-sequencing (RNA-seq) analysis in order to quantify the transcriptomic landscape. As demonstrated by volcano plot in Fig. 6a, DR3 stimulation of ILC2s resulted in differential modulation of 579 genes (305

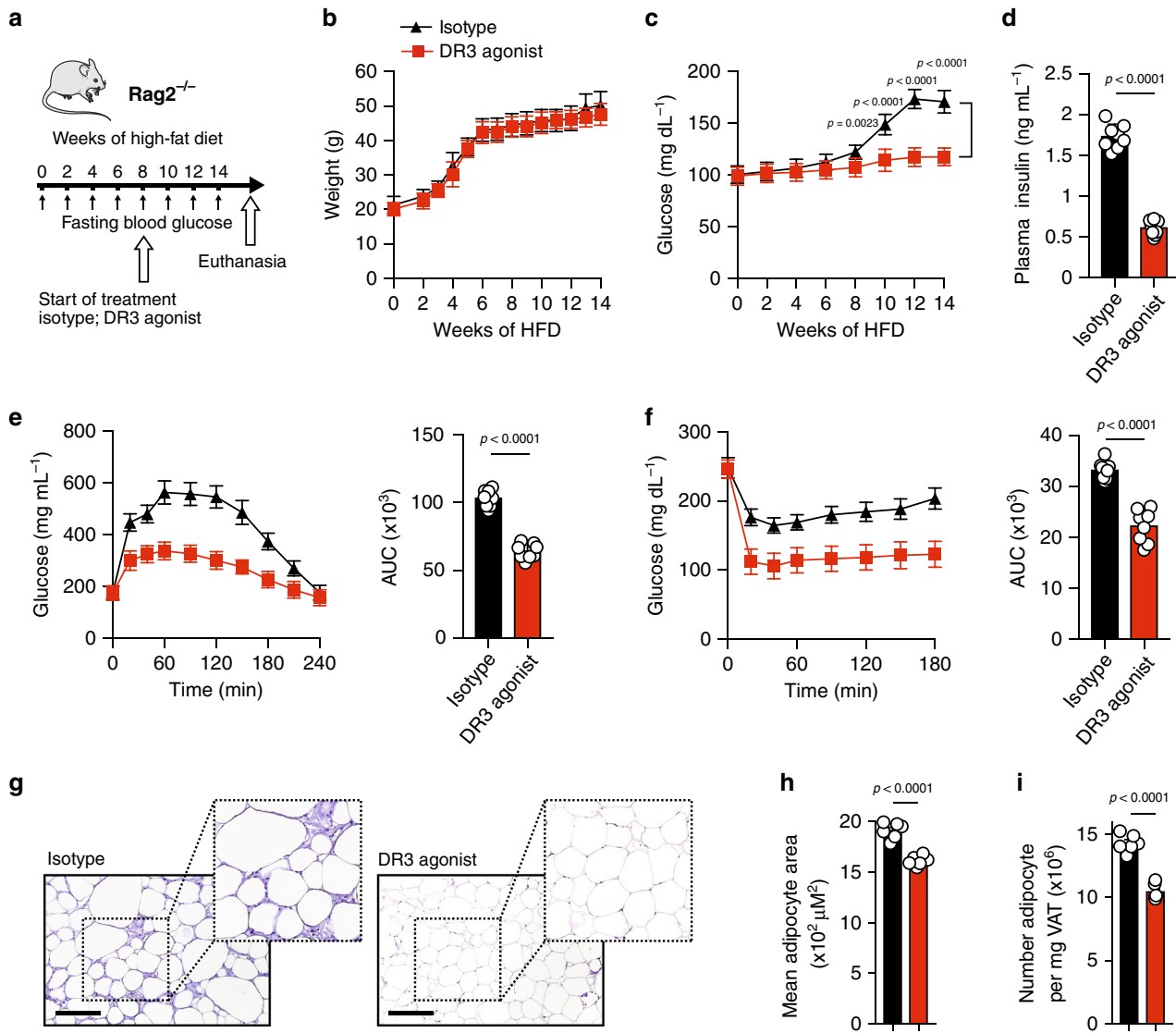

**Fig. 4 Therapeutic DR3 treatment ameliorates established type 2 diabetes in *Rag2*⁻/⁻ mice. a** A cohort of *Rag2*⁻/⁻ mice were placed on HFD for 14 weeks. After 8 weeks of diet, mice were subjected to intraperitoneal treatments with either DR3 agonist (1 mg/mouse) or isotype control once every four days as indicated in the timeline, $n = 8$ mice. **b** Total weights were measured once a week, and **c** the fasting blood glucose levels were measured once every 2 weeks for a period of 14 weeks. **d** Plasma insulin concentrations were determined using ELISA. **e** Glucose tolerance test and **f** insulin tolerance test were performed at the end of the treatments. **g** Hematoxylin and eosin-stained epididymal adipose tissue sections (×400, scale bars, 100 µm). The mean adipocyte area (**h**) and numbers (**i**) in VAT were quantified at the end of 14 weeks. The area under the curve (AUC) was calculated for each cohort. Error bars are representative of the mean ± SD. Statistical analysis, one-way ANOVA (**b**, **c**), two-way ANOVA (**e**, **f**), two-tailed student's *t*-test (**d**, **h**, **i**). Mouse image provided with permission from Servier Medical Art.

downregulated, 274 upregulated, $p < 0.05$, 1.5FC), further depicted as a heat map (Fig. 6b). Moreover, the relevant cytokine and cytokine receptor genes were correlated based on their modulation in response to DR3 agonist (Fig. 6c). Upregulation of ILC2 relevant activation cytokines, such as *Il5*, *Il6*, *Il9*, and *Csf2* (GMC-SF) at a transcriptional level is consistent with our previous ex vivo data (Fig. 1d). Furthermore, DR3 stimulation modulated the gene expression of various transcriptional factors, such as *Nfkb2*, *Nfkbia*, and *Nfkbie* (Fig. 6d). We further analyzed the pathways underlying the gene signature regulated by DR3 engagement on ILC2s using the Ingenuity Pathway Analysis (IPA) tool. IPA revealed that genes critically involved in the NF-κB pathway were significantly (*p*-value = $1.41 \times 10^{-6}$) enriched in response to DR3 stimulation when compared to the isotype treated control group (Fig. 6e). More specifically, the non-canonical NF-κB pathway was significantly enriched in activated

ILC2s. We found that activated VAT-derived ILC2s stimulated with DR3 demonstrate significant upregulation of key genes associated with the non-canonical NF-κB pathway, including *Map3k14* (*protein* NF-κB inducing kinase), *Nfkb2* (p52) and *Relb*, whereas canonical NF-κB genes *Nfkb1* and *Rela* (p65) were unchanged at a transcriptional level (Fig. 6e). In order to validate the RNA-seq results, we next examined the protein expression levels of p65 and p52 in the canonical and non-canonical NF-κB pathways, respectively. Activated ILC2s were again purified from VAT of C57BL/6 mice after 3 doses of i.p. IL33 and treated with agonist DR3 antibody for 24 h ex vivo followed by intracellular flow cytometry for p52 and p65. In agreement with our transcriptional findings, p52 was upregulated at a protein level after DR3 stimulation over isotype treated controls (Fig. 6f), whereas p65 appeared to be unaffected by DR3 signaling in activated ILC2s (Fig. 6g). Altogether, these results demonstrate that DR3

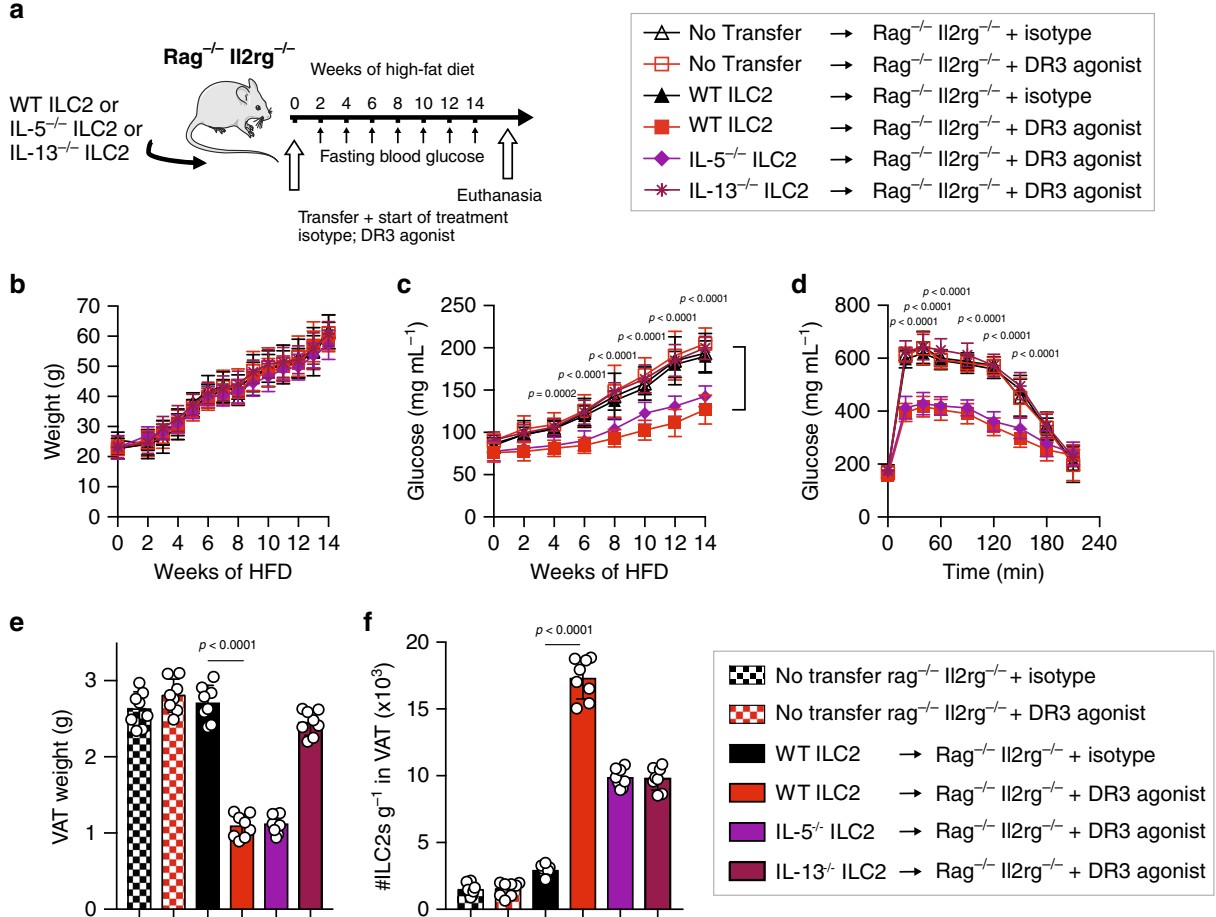

**Fig. 5 The therapeutic effects of DR3 are dependent on expression on ILC2s and IL-13. a** A cohort of $Rag^{-/-}$ $Il2rg^{-/-}$ mice were adoptively transferred with ILC2s from either WT, $Il5^{-/-}$ or $Il13^{-/-}$ mice, $n = 8$ mice. Moreover, two additional mice cohorts did not receive any ILC2s. Subsequently, mice were placed on HFD for a period of 14 weeks and were intraperitonially treated with either DR3 agonist or isotype control as shown in the timeline. **b** Total weights were measured once a week, and **c** fasting blood glucose levels were measured once every two weeks for a period of 14 weeks. **d** Glucose tolerance test was performed on the 14th week. **e** the weight of VAT was determined, and **f** the number of ILC2s per gram of VAT was quantified after 14 weeks. Error bars are the mean ± SD. Statistical analysis, one-way ANOVA (**b**–**d**), two-tailed student's $t$-test (**e**, **f**); ns: non-significant. Mouse image provided with permission from Servier Medical Art.

engagement serves as a co-stimulatory signal for secretion of type 2 cytokines and selectively induces the non-canonical NF-κB signaling pathway via *NIK*, *Nfkb2*, and *Relb* in VAT-derived activated ILC2s.

As our ex vivo results (Fig. 1d) suggest that DR3 is capable of activating naïve ILC2s, we next explored the molecular mechanisms of DR3-dependent activation by isolating naïve ILC2s from WT mice and culturing them in presence of DR3 agonist or isotype control (5 μg/mL) for 24 h ex vivo. We then performed RNA-seq analysis in order to assess the presence and quantity of RNA transcripts. DR3 stimulation of naïve ILC2s resulted in differential modulation of 363 genes (137 downregulated, 226 upregulated, $p < 0.05$, 1.5FC), as shown in Supplementary Fig. 3a, b. The relevant cytokine and cytokine receptor genes were correlated based on their modulation in response to DR3 agonist (Supplementary Fig. 3c). Upregulation of *Il5*, *Il13*, and *Csf2* (GMC-SF) at a transcriptional level is consistent our previous ex vivo (Fig. 1d) data. Furthermore, DR3 stimulation resulted in modulation of various transcriptional factors, such as *gata3*, *Nfkb2*, *Nfkbia*, and *Nfkbie* (Supplementary Fig. 3d). In order to distinguish signaling between the canonical and non-canonical NF-κB pathways, we next examined the protein expression levels

of p52 and p65 in the two NF-κB pathways, respectively. In agreement with our transcriptional findings in activated ILC2s, p52 was upregulated at a protein level after DR3 stimulation of naïve ILC2s (Supplementary Fig. 3e). Interestingly, p65 was similarly upregulated by DR3 engagement (Supplementary Fig. 3f) on naïve VAT-derived ILC2s. We further analyzed the two aforementioned p52 and p65 NF-κB pathways in mice fed HFD or NCD for 14 weeks in vivo (Supplementary Fig. 4). Importantly, the DR3 downstream signaling was not impaired in naïve nor activated ILC2s of the obese model, suggesting that DR3 remains a viable therapeutic target in the context of established metabolic syndrome. Altogether, these results demonstrate that DR3 engagement can induce signaling through the engagement of the canonical and non-canonical NF-κB in naïve ILC2s, and in activated ILC2s can promote the non-canonical pathway.

**DR3 engagement on human ILC2s induces type 2 cytokines**. In order to determine the translation potential of our findings, we next investigated the conditions under which human ILC2s express DR3. We FACS purified ILC2s as lineage⁻CD45⁺ CD127⁺ and CRTH2⁺ cells (Fig. 7a) from human PBMCs of healthy donors and cultured them with recombinant human (rh)

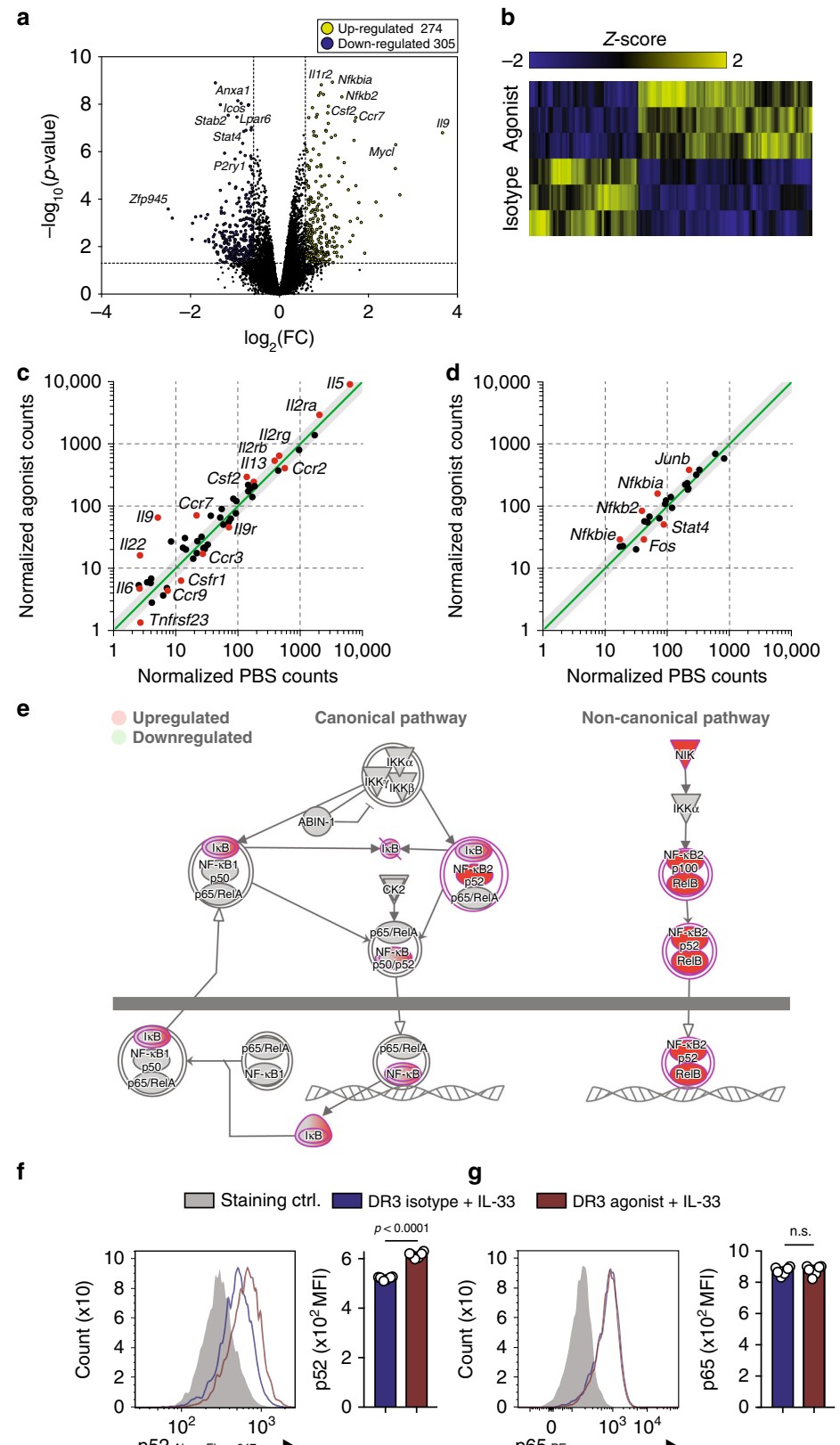

IL-2 and rhIL-7. The human ILC2s were subsequently stimulated ex vivo with rhIL-33, and DR3 expression was analyzed over time at 12, 24, and 48 h by flow cytometry (Fig. 7b). We found that both naïve and IL-33-activated human ILC2s express DR3 (Fig. 7c). Moreover, we observed that DR3 expression increased over time and reached statistical significance after 24 h of ex vivo

rhIL-33 stimulation (Fig. 7c). We next asked whether DR3 engagement can induce and/or enhance the effector function of naïve and activated human ILC2s. Purified peripheral blood ILC2s from healthy donors were cultured with rhIL-2, rhIL-7 in the presence of rhIL-33 and/or known ligand TL1A-L (Fc-Fusion) or vehicle control (PBS). After 48 h, the supernatant was

**Fig. 6 DR3 signaling induces non-canonical NF-κB pathway in activated VAT-derived ILC2s.** In vivo activated VAT-derived murine ILC2s (aILC2s) were cultured in presence of recombinant mouse (rm) IL-2, rmIL-7 and rmIL-33 and stimulated with DR3 agonist (5 μg/mL) or isotype control for 24 h. Total RNA was isolated and sequenced. **a** Volcano plot comparison representing whole transcriptome gene expression of sorted WT ILC2s treated with either isotype control or DR3 agonist (5 μg/mL) for 24 h ex vivo. Differentially expressed genes (described as statistically significant adjusted $p$-value < 0.05) with changes of at least 1.5-fold change (FC) are shown in yellow (upregulated) and blue (downregulated). Relevant differentially expressed genes are identified. **b** Heat plot of all differentially expressed genes. **c** Selected cytokine and cytokine receptor genes and transcription factors (**d**) plotted as the normalized counts in isotype-treated cohort compared to DR3 stimulated cohort. Notable ILC2 related genes are labeled and highlighted in red. Gray area represents region of 1.5-fold change in gene expression. **e** Upregulated (red) and downregulated (green) genes in the canonical and non-canonical NF-κB pathways. **f** Representative expression of NF-κB p52 (**f**) and NF-κB p65 (**g**) in isolated ILC2s from mice challenged with IL-33 and cultured ex vivo for 24 hrs with DR3 agonist (red) or isotype control (blue), $n = 6$ mice. The staining FMO control is shown as grey. The corresponding quantification are presented as Mean Fluorescence Intensity (MFI), and the error bars denote the mean ± SD. Statistical analysis, two-tailed student's $t$-test (**f**, **g**); n.s.: $p$-value of <0.05 was considered as non-significant.

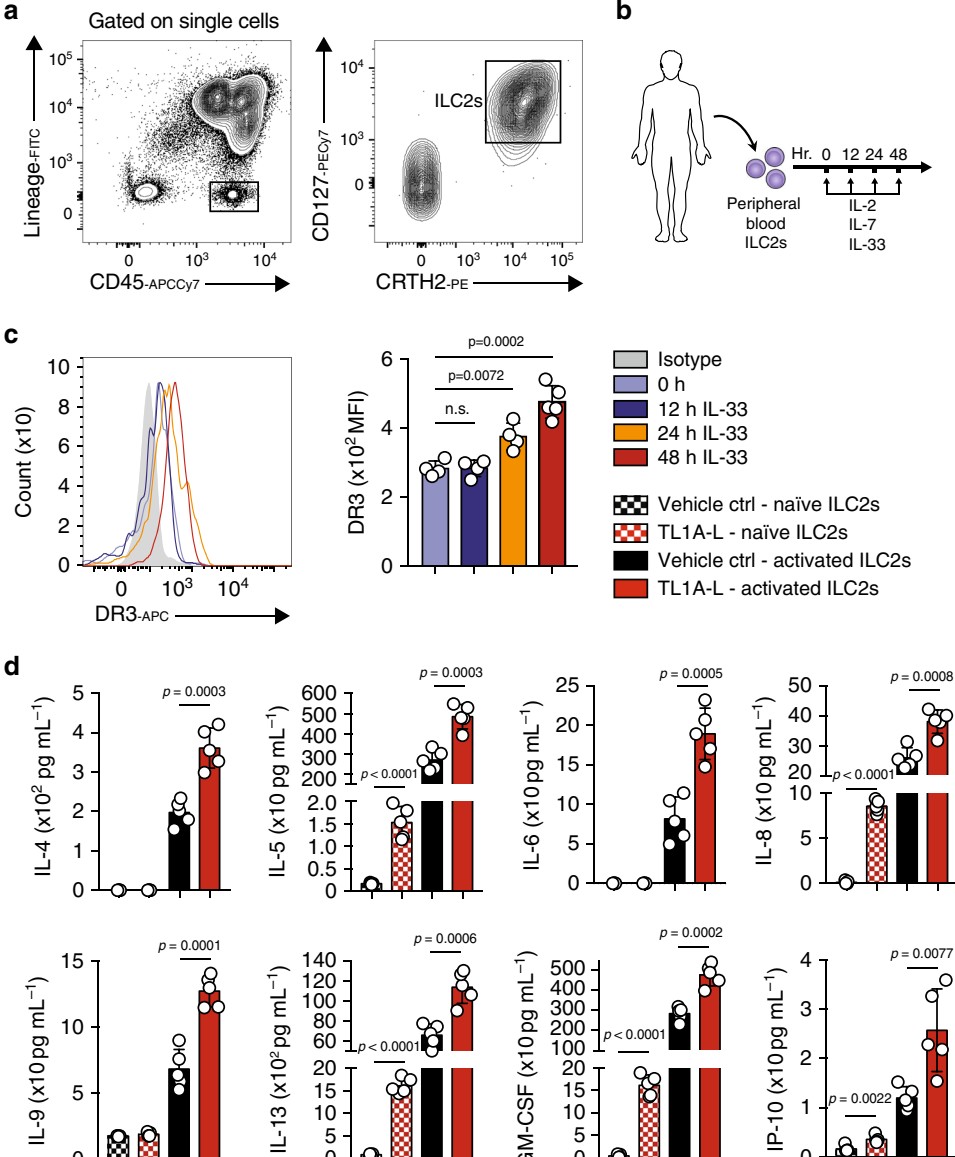

**Fig. 7 Human ILC2s express DR3, and this expression is inducible by IL-33. a** The gating strategy of Lin⁻CD45⁺CD127⁺CRTH2⁺ human ILC2 cells. **b** Human peripheral-blood ILC2s were freshly sorted and cultured with 10 ng/mL of recombinant human (rh) IL-2 and rhIL-7 in presence or absence of rhIL-33 (10 ng/mL) for 12, 24, and 48 h. Freshly isolated ILC2 at 0 h and ex vivo activated ILC2s were analyzed by flow cytometry as indicated in the scheme, and (**c**) the kinetics of DR3 induction by IL-33 is shown. **d** The levels of IL-5, IL-6, IL-8, IL-9, IL-13, and GM-CSF in the culture supernatants were measured by Luminex after 48 h of stimulation. Data representative of five individual blood donors, $n = 5$. Error bars are the mean ± SD. Statistical analysis, one-way ANOVA (**c**) or two-tailed student's $t$-test (**d**); n.s.: $p$-value of <0.05 was considered as non-significant. Human image provided with permission from Servier Medical Art.

collected, and cytokine levels were measured via Luminex assay. Consistent with our murine ILC2s observations, IL-5, IL-13, and GM-CSF levels were augmented by DR3 engagement on naïve human ILC2s, whereas IL-6 and IL-9 levels were unaffected (Fig. 7d). DR3 engagement on activated ILC2s dramatically increased the secretion levels for all the aforementioned cytokines, including IL-6 and IL-9 (Fig. 7d). Overall, these results demonstrate that naïve and activated human ILC2s express DR3, and DR3 engagement on human ILC2s can provide both stimulatory and co-stimulatory signals.

## Discussion

In the present study, we establish the mechanism, signaling pathways, and therapeutic potential of DR3 engagement on VAT-derived ILC2s in context of insulin-resistance and T2DM. We ascertain that DR3, a co-stimulatory member of the TNFR superfamily, is expressed on both human and mouse VAT-derived ILC2s, and this expression is inducible by IL-33. Strikingly, DR3 engagement induces type 2 cytokine production in both naïve and activated ILC2s. Moreover, we demonstrate that DR3 engagement induces both the canonical and/or non-canonical NF-κB pathways in naïve and activated ILC2s as evidenced by upregulation of p52, p65, NIK (encoded by Map3k14), Nfkb2, and Relb. Using a DR3 specific agonist antibody, we establish that DR3 engagement on ILC2s both protects against the development of T2DM and is capable of reversing already established T2DM. The therapeutic efficacy of DR3 engagement on ILC2s was consistent in mice from two different genetic backgrounds (BALB/c and C57BL/6J). Future studies should also consider the C57BL/6N mouse model, which has altered metabolic handling[33]. We further demonstrate the potent therapeutic effects of DR3 agonist are IL-13 dependent. Our results emphasize the critical role of DR3 as a modulator of ILC2 effector functions, provide new insights into the role of DR3 in VAT-derived ILC2s and establish DR3 agonist treatment as a promising novel therapeutic avenue to prevent and treat T2DM.

It is established that in both obese animal models and humans, low-grade inflammation exists in the adipose tissue[34]. This can be caused by a leaky gut and release of microbial products or other stimulants that promotes proinflammatory cells such as M1 macrophages and T effector cells[35,36]. On the other hand, cells such as endothelial cells and stromal cells produce IL-33 to maintain the homeostasis of ST2+ anti-inflammatory cells such as ILC2s and Tregs[26,37,38]. IL-33 upregulates numerous pathways including MyD88/IRAK/TRAF6, ERK1/2, JNK, p38, and PI3K/AKT, and all these pathways have been shown to influence expression of different co-stimulatory receptors[39]. However, the contribution of each pathway to induction of various co-stimulatory receptors remains to be elucidated. In obese patients with T2DM, the meticulous balance between pro-inflammatory and anti-inflammatory homeostasis dangerously tips toward active chronic inflammation[40]. Pro-inflammatory cytokines, including type 1 and type 2 interferons possess the ability to suppress ILC2s[41–44]. The deleterious role of type 1 interferons in T2DM was suggested several years ago as insulin resistance developed in human subjects injected with IFN-α[45]. Furthermore, the risk of developing metabolic disorders, such as T2DM, has been established in autoimmune diseases that are associated with type 1 interferons such as systemic lupus erythematosus and psoriasis[46,47]. Several groups have suggested that the source of IFN-α is mainly TLR9 activated adipose tissue pDC most likely as a result of a leaky gut[48]. Our group recently reported that production of pDC-derived IFN-α restricts expansion, survival and effector function of ILC2s[41]. Moreover, several contemporaneous studies highlighted that IFN-β and IFN-γ,

which are also abundant in adipose tissue of obese patients, exhibit similar inhibitory effects on ILC2s[41–44]. The type 1 environment in adipose tissue of obese patients is detrimental to survival of ILC2s and contributes to the low number and insufficient activation of VAT ILC2s. Although tissue-resident ILC2s do not traffic to other tissues, there has been several lines of work that show BM derived ILC2s have the ability to traffic and replenish various tissues[49–51]. We believe due to the inhibitory cytokine milieu, the new flux of naïve ILC2s from the bone marrow—before they get a chance to get sufficiently activated by endothelial-derived IL-33—are profoundly suppressed and do not survive in VAT of obese patients. As a result, our group and others have now reported low numbers of adipose-resident ILC2s in murine models of HFD and obese patients[6,43]. Therefore, we believe targeting DR3 can address the dire need for a therapeutic agent that can stimulate the recently arrived naïve and insufficiently activated tissue-resident ILC2s.

We herein demonstrate that DR3 stimulation promotes the effector functions of both naïve and activated ILC2s through the canonical and non-canonical NF-κB pathways. Our results suggest that DR3 engagement in naïve ILC2s induces upregulation of both canonical and non-canonical NF-κB pathways as evidenced by p52 and p65 staining. Since IL-33 robustly upregulates p65 protein, the engagement of DR3 was only able to upregulate p52 protein in activated ILC2s. Activation of both canonical and non-canonical NF-κB pathways by DR3 signaling can potentially produce a synergistic effect[52], which further ensures potent and sufficient activation of ILC2s to ameliorate T2DM. However, future studies will be necessary to determine the relative contribution of each pathway. We believe this therapeutic approach would be superior to the recently published results by our group targeting other co-stimulatory molecules such as GITR, which only stimulates the activated, but not naïve ILC2s[28]. Interestingly, the hepatic weight was not altered by the engagement of neither DR3 nor GITR on VAT ILC2s. Although we previously observed an increase in the lean mass by GITR-dependent stimulation of VAT ILC2s, future studies are needed to explore the effect of DR3 agonistic treatment on the skeletal muscles and lean mass.

ILC2s have been documented to modulate the type 2 immune responses through copious secretion of key type 2 cytokines such as IL-5 and IL-13. These cytokines have been shown to play an important role in recruitment and accumulation of eosinophils, and thus indirect maintenance of AAMs in the adipose tissue[28]. In this study, we also described DR3-dependent amelioration of T2DM and metabolic syndrome, coinciding with beiging of the adipose tissue, a higher metabolic rate and the induction VAT-associated CD206+ AAM phenotype. Previous studies have shown that metabolic syndrome is exacerbated in IL-5 or IL-13 deficient mice that were placed on a HFD. In accord, mice overexpressing IL-5 or treated with IL-13 have been reported to be leaner and more glucose tolerant[53]. We further performed studies, utilizing the adoptive transfer of ILC2s in alymphoid recipients to show that engagement of DR3 on ILC2s is sufficient to ameliorate T2DM. Although DR3 is expressed on a variety of cells[54–56], results of the aforementioned experiments were proof of concept for the ability of ILC2s to dampen the phenotypes associated with T2DM. Successful future clinical studies must consider the overall expression pattern of DR3 on lymphoid and non-lymphoid tissues described above. Moreover, since pulmonary ILC2s have been shown to exacerbate airway inflammation in allergic asthma, future clinical studies must aim to modulate the activity of ILC2s in different tissues[57,58]. For instance, factors such as modes of administrations, hydrophobic modifications, and delivery options should be considered to avoid any potential adverse side effects such as lung inflammation.

Recent studies have highlighted the critical interplays between ILC2s and their respective physical tissue niches during homeostasis and inflammation. Type 2 cytokines produced by ILC2s may exert a direct impact on somatic cells such as adipocytes. We demonstrated that DR3 engagement improves local hyperplasia and hypertrophy in VAT. Nevertheless, the interplay between the stroma and adipose-resident immune cells remains an area of immense interest. Recently, a study showed that white adipose tissue-resident multipotent stromal cells (WAT-MSCs) support ICAM-1-mediated proliferation and activation of ILC2s[38]. In the lungs, adventitial stromal cells (ASCs) reportedly regulate local ILC2 function and expansion[59]. Future studies are needed to explore the tissue-specific programing of ILC2s in various organs and tissues. A recent study showed that DR3 is expressed on intestinal ILCs, and another group has reported that intestinal ILC2s contribute to T2DM and obesity in mice[60,61]. Here we demonstrate that DR3 agonistic treatment is an effective method for amelioration of T2DM regardless of the differences among various tissue resident ILC2s.

We further observed that ILC2-derived IL-13, but not IL-5, plays a significant role in reducing the blood glucose concentration and increasing the sensitivity to insulin in the above-mentioned adoptive transfer experiments. Interestingly, previous studies have reported a preventive effect of IL-13 on the onset of diabetes in non-obese diabetic mice (NOD mice)[62], providing evidence that IL-13 downregulates the immune-inflammatory diabetogenic pathways in agreement with our findings. Furthermore, recent studies on prediabetic subjects suggest that the level of IL-13 is negatively associated with risk of type 2 diabetes and can be considered as an anti-inflammatory marker for incidence of T2DM and insulin resistance[62,63]. Brahimaj et al. evaluated 26 inflammatory biomarkers in a longitudinal analysis of approximately a thousand subjects from the Rotterdam Study. In this epidemiological study, plasma IL-13 was inversely linked with progression from normoglycemia to pre-diabetes, incidence of type 2 diabetes, and initiation of insulin therapy[63].

We explored the relevance of DR3-dependent stimulation of human ILC2s. We demonstrated that expression of DR3 is inducible on human ILC2s by IL-33 in a time-dependent manner. Consistent with our murine results, DR3 engagement induces type 2 cytokine production in both naïve and activated human ILC2s. Overall, our results suggest that the DR3 axis provides the necessary signals for the activation of human ILC2s. In clinical settings, incoming type 2 diabetes patients have an established insulin-resistance. Therefore, it is important to not only prevent, but also to reverse their established metabolic syndrome. Importantly, we demonstrated that DR3 engagement asserts not only a preventive role against the onset of insulin-resistance, but also a therapeutic effect in models with previously established metabolic syndrome. Collectively, these results suggest DR3 agonistic treatment as a promising therapeutic avenue to reverse insulin-resistance in type 2 diabetes patients.

## Methods

**Mice**. C57BL/6J (stock number: 000664), BALB/cByJ (stock number: 000651), RAG2 deficient (C.B6(Cg)-Rag2tm1.1Cgn/J) (stock number: 008448), RAG2-, IL-2Rg-deficient (C;129S4-Rag2tm1.1Flv Il2rgtm1.1Flv/J) (stock number: 014593) were purchased from the Jackson Laboratory (Bar Harbor, Maine). BALB/c IL-5 deficient mice (11 generations derived from IL-5 deficient C57BL/6), and BALB/c IL-13 deficient mice was obtained from A. McKenzie (MRC Laboratory of Molecular Biology, Cambridge, UK)[64,65]. All mice were bred in our animal facility at the Keck School of Medicine, University of Southern California (USC). Mice were maintained at macroenvironmental temperature of 21–22 °C, humidity (48–52%), in a conventional 12:12 light/dark cycle with lights on at 6:00 a.m. and off at 6:00 p.m. In all, 4–8 weeks-old age and sex matched male mice were used in the studies. All animal studies were approved by the USC institutional Animal Care and Use Committee and conducted in accordance with the USC Department of Animal Resources' guidelines.

**Human subjects**. Human blood samples were obtained from male and female healthy donors (Age 18–65). No specific selection criteria was applied for healthy donors. All human studies were approved by USC Institutional review board and conducted in accordance to the principles of the Declaration of Helsinki. Informed consent was obtained from all donors according to our approved IRB protocols.

**Diet-induced obesity and in vivo treatments**. At indicated times, the mice were fed a HFD (Rodent diet with 60 kcal% Fat, D12492i) from Research Diets Inc. (New Brunswick, New Jersey) for the indicated times[28]. Briefly, all other mice were fed a normal chow diet (NCD). For in vivo experiments investigating the effect of DR3 engagement, DR3 agonist (1 mg/mouse, BioLegend, San Diego, CA, clone 4C12), or the monoclonal antibody Armenian Hamster IgG Isotype Ctrl (BioLegend) (1 mg/mouse) was administered intraperitoneally every 4 days from the indicated start of treatment until termination of the experiment.

**In vivo metabolic phenotyping**. In order to measure weight and fasting blood glucose levels, mice were fasted overnight (~14 to 16 h), weighed. Glucose was measured by a drop of blood using a glucometer (Contour®Next EZ, Bayer, Leverkusen, Germany) collecting a drop of blood every two weeks. For intraperitoneal glucose tolerance tests (ip-GTT), mice were fasted overnight (~16 h), weighed and injected with 2 g/kg 20% D-glucose (Sigma Aldrich) solution intraperitoneally. Blood glucose values were measured for each mouse by collecting a drop of blood before injection and at 20, 40, 60, 90, 120, 150, 180, 210, and 240 min post-injection. For insulin tolerance tests (ITT), mice were fasted for 5 h, weighed and injected with 0.5U/kg human insulin (Novolin®, Novo Nordisk®, Bagsværd, Denmark) diluted in Sodium Chloride Solution 0.9% w/v (Azer Scientific, Morgantown, Pennsylvania) solution intraperitoneally. Blood glucose values were measured for each mouse by collecting a drop of blood before injection and at 20, 40, 60, 90, 120, 150, 180, 210, and 240 min post injection. For glucose and insulin tolerance tests, the data were analyzed by quantifying the area under the curve (AUC) for each group of mice. When indicated, blood was collected by cardiac puncture and plasma insulin levels were measured using the ultra-sensitive mouse insulin ELISA Kit (Crystal Chem High Performance Assays).

**Murine ILC2 isolation and ex vivo stimulation**. Murine VAT and human peripheral ILC2s were isolated to >95% purity using the FACS Aria III cell sorter. For in vivo stimulation of murine VAT ILC2s, carrier free rm-IL-33 (Biolegend, San Diego, CA, 1 µg/mouse in 200 µL) was administered intraperitoneally to mice on three consecutive days. On day 4, murine ILC2s were isolated based on the lack of expression of classical lineage markers (CD3ε to mice on three consecutive days. On day TCRγδ and FCεRI) and positive expression of CD45, ST2, and CD117. Isolated ILC2s were stimulated ($5 \times 10^4$/mL) with rm-IL-2 (10 ng/mL) and rm-IL-7 (10 ng/mL) for 48 h at 37 °C in presence of DR3 agonist (5 µg/mL) from BioLegend, San Diego, CA (clone 4C12, or the monoclonal antibody Armenian Hamster IgG Isotype Ctrl (BioLegend). For adoptive transfer experiments, $2.5 \times 10^5$ purified ILC2s were adoptively transferred intravenously in 200 µL PBS into the recipients at the start of the indicated treatment.

**Human ILC2 isolation and ex vivo stimulation**. For human ILC2s from blood, peripheral blood mononuclear cells (PBMCs) were first isolated from fresh human blood by diluting the blood 1:1 in PBS and adding to SepMate™-50 separation tubes (STEMCELL Technologies Inc, Vancouver, Canada) prefilled with 15 mL Lymphoprep™ each (Axis-Shield, Oslo, Norway) and centrifugation at $1200 \times g$ for 15 mins. Human ILC2s were then isolated by cell sorting based on the lack of expression of classical lineage markers (CD3, CD5, CD14, CD16, CD19, CD20, CD56, CD235a, CD1a, CD123) and expression of CD45, CRTH2 and CD127. Purified human ILC2s were stimulated ($5 \times 10^4$ per mL) with recombinant human (rh)-IL-2 (20 ng/mL), rh-IL-7 (20 ng/mL) for the indicated times at 37 °C in presence or absence of rh-IL-33 (20 ng/mL) and/or TL1A-L (Fc-Fusion) from AB Bioscience (P7042F).

**Supernatant cytokine measurement**. ELISA for mouse IL-5 and IL-13 were purchased from ThermoFisher Scientific and the level of cytokines were measured according to the manufacturer's instructions. Other cytokines were measured by multiplexed fluorescent bead-based immunoassay detection (MILLIPLEX® MAP system, Millipore Corporation, Missouri U.S.A.) according to the manufacturer's instructions, using a 32-plex (MCYTMAG70KPX32) and 41-plex (HCYTMAG-60K-PX41) Millipore Human Cytokine panel kits. For each assay, the curve was performed using various concentrations of the cytokine standards assayed in the same manner and analyzed using MasterPlex2012 software (Hitachi Solutions America, Ltd.), as extensively described by our group[66,67].

**Tissue preparation and Flow cytometry**. Epididymal adipose tissue used as representative VAT was collected at the indicated times after transcardial perfusion to clear organs of red blood cells. VAT was processed to single-cell suspensions as demonstrated before[68]. Stained cells were analyzed on FACSCanto II and/or FACSARIA III systems (Becton Dickinson) and the data were analyzed with FlowJo version 10 software (TreeStar, Ashland, Oregon). Briefly, the following

mouse antibodies were used: biotinylated anti-mouse lineage CD3e (145-2C11; 1/200), CD5 (53-7.3; 1/200), CD45R (RA3-6B2; 1/200), Gr-1 (RB6-8C5; 1/200), CD11c (N418), CD11b (M1/70; 1/200), Ter119 (TER-119; 1/200), Fcεc F (MAR-1) (BioLegend; 1/200) and TCR-gd (eBioGL3;1/200) (eBioscience), Streptavidin-FITC (1/500), PE-Cy7 anti-mouse CD127 (A7R34; 1/200), APCCy7 anti-mouse CD45 (30-F11; 1/200), PE anti-mouse F4/80 (BM8; 1/300), FITC anti-mouse CD206 (C068C2; 1/300), APC/Cy7 anti-mouse CD11c (N418; 1/300), APC anti-mouse DR3 (4C12; 1/200) and corresponding isotype control Armenian Hamster IgG (HTK888; 1/200) were purchased from BioLegend. Alexa 647 anti-mouse NFκB p52 (C5) was purchased from Santa Cruz Biotechnology. PE anti-human/ mouse RelA NFκB p65 (IC5078P; 1/200) and PE IgG2B (133303; 1/200) was purchased from R&D Systems. eFluor450 anti-mouse CD11b (M1/70; 1/200) was purchased from eBioscience. The following human antibodies were used: FITC- Lineage that includes mixture of CD3, CD14, CD16, CD19, CD20, CD56 (348801; 1/100), FITC-CD235a (HI264; 1/500), FITC-FCeRIa (AER-37; 1/100), FITC-CD1a (HI149;1/100), FITC-CD123 (6H6; 1/100), FITC-CD5 (L17F12; 1/100), APCCy7-CD45 (HI30; 1/ 100), PE-CD294 (CRTH2) (BM16; 1/100), PE/Cy7-CD127 (A019D5; 1/100), APC - IgG2ak (MOPC-173; 1/200) were pur- chased from BioLegend. APC anti-human DR3 (JD3; 1/200) was purchased from Miltenyi Biotec.

**RNA sequencing (RNA-seq) and data analysis**. Freshly isolated ILC2s after 3 i.p. injections of 1 μg rm-IL33 were stimulated ($5 \times 10^4$/mL) with rm-IL-2 (10 ng/mL) and rm-IL-7 (10 ng/mL) at 37 °C with DR3 agonist (5 μg/mL) or the isotype control for 24 h. Total RNA was isolated using MicroRNAeasy (Qiagen, Valencia, California). In all, 10 ng of input RNA was used to produce cDNA for downstream library preparation. Samples were sequenced on a NextSeq 500 (Illumina) system. Raw reads were aligned, normalized and further analyzed using Partek® Genomics Suite® software, version 7.0 Copyright ©; Partek Inc., St Louis, MO, USA. Pathway analysis was performed using the Qiagen Ingenuity Pathway Analysis (IPA) software.

**Real-time PCR**. The total RNA was prepared via the RNeasy lipid tissue kit (QIAGEN, Hilden, Germany) following the manufacturer's instructions and quantitative PCR was performed on a LightCycler 2.0 (Roche Diagnostics, Man- nheim, Germany). The relative mRNA levels were examined by comparison with a reference gene using the delta-delta CT method, as described before[28,69,70]. Briefly, the sequences of the gene-specific primers are: Cidea (forward primer) CGAGT TTCAAACC-ATGACCGAAGTAGCC, Cidea (reverse primer) CTTACTACCCG GTGTCCATTTCTGTCCC, Cox7a (forward primer) CTCTTCCAGGCCGACAA TGACCTC, Cox7a (reverse primer) G-CCCAGGCCCAAGCAGTATAAGCA, Dio2 (forward primer) TACAAACAGGTTAAACTGGGT-GAAGATGCTC, Dio2 (reverse primer) GAGCCTCATCAATGTATACCAACAGGAAGTC, Hprt (for- ward primer) GCTGGTGAAAAGGACCTC, Hprt (reverse primer) CACAGG ACTAGAACA-CCT, Prdm16 (forward primer) TCTACATTCCTGAAGACATTC CAATCCCACCA, Prdm16 (reverse primer) TGTATCCGTCAGCATCTCCCATC CAAAGTC, Pgc1a (forward primer) AA-GACAGGTGCCTTCAGTTCACTCTCA G, Pgc1a (reverse primer) AGCAGCACACTCTATGT-CACTCCATACAG, Ucp1 (forward primer) GATGGTGAACCCGACAACTTCCGAAGTG, Ucp1 (reverse primer) TTCACCTTGGATCTGAAGGCGGACTTTGG, Il-33 (forward primer) GGTGAACATGAGTCCCATCA, Il-33 (reverse primer) CGTCACCCCTTTGAA GCTC, Nd1 (forward primer) CAGGATGAGCCTCAAACTCC, Nd1 (reverser primer) GGTCAGGCTGGCAGAAGTAA, Nd2 (forward primer) AGGGATCCCA CTGCACATAG, Nd2 (reverser primer) CCTATGTGGGCAATTGATGA, Th (forward primer) TGTTGGCTGACCGCACAT, Th (reverser primer) GCCCC CAGAGATGCAAGTC, Cytb (forward primer) ACGTCCTTCCATGAGGACAA, Cytb (reverse primer) GAGGTGAACGATTGCTAGGG, Cox1a (forward primer) CTTTTTATCCTCCCAGGATTTGG, Cox1a (reverser primer) GCTAAATACT TTGACACCGG, Atp6 (forward primer) AATTACAGGCTTCCGACACAAAC, Atp6 (reverser primer) TGGAATTAGTGAAATTGGAGTTCCT.

**Histologic analysis**. Isolated samples were fixed with 4% paraformaldehyde in PBS. After fixation, tissues were embedded in paraffin, before cutting into 4 μm sections and staining with hematoxylin and eosin (H&E). Stained sections were acquired using a Leica DME microscope and Leica ICC50HD camera (Leica, Wetzlar, Germany) and analyzed using Leica LAS EZ software. Adipocyte numbers and area were quantified as formerly demonstrated[12,71].

**Statistical analysis**. Experiments were repeated at least three times ($n = 4–8$ each) and data are shown as the representative of three independent experiments. Two- tailed student $t$-test, a one-way or two-way analysis of variance (ANOVA) followed by Tukey post hoc tests were used to determine statistical significance using Prism Software (GraphPad Software Inc.). Error bars represent standard deviation (SD) of the mean.

**Reporting summary**. Further information on research design is available in the Nature Research Reporting Summary linked to this article.

## Data availability
Sequence data that support the findings of this study (Fig. 6 and Supplementary Fig. 3) have been deposited in Genbank with the primary accession code GSE154823. Source data are provided with this paper.

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

## Acknowledgements

This article was financially supported by National Institutes of Health Public Health Service grants R01 ES025786, R01 ES021801, R01 HL144790, R01 HL151493, R21 AI109059 (O.A.). We are grateful to USC Libraries Bioinformatics Service for assisting with data analysis, in particular Dr. Yibu Chen. The bioinformatics software and computing resources used in the analysis are funded by the USC Office of Research and the Norris Medical Library.

## Author contributions

P.S.J. designed, performed and analyzed all experiments and wrote the paper. B.P.H., L.G., E.H., D.G.H., J.P., R.L., G.L. helped perform experiments and provided animal husbandry for experiments. P.S. assisted with design and interpretation of data. O.A. supervised, designed the experiments, conceptualized, interpreted the data and finalized the paper. All authors helped with reviewing and editing the paper.

## Competing interests

The authors declare no competing interests.
