## [Peer Review File · Nature Communications]

Reviewers' Comments:

Reviewer #1:

Remarks to the Author:

Shafiei-Jahani et al. examined the role of DR3 on ILC2 and demonstrated in this paper that stimulation of ILC2 through DR3 by agonistic antibody protected mice from high-fat diet (HFD)-induced hyperglycemia and insulin resistance and reversed previously established insulin-resistance in an IL-13-dependent manner. The authors also showed that DR3 engagement enhanced non-canonical NK-kB pathway to enhance ILC2 functions. The authors also demonstrated that human ILC2 derived from peripheral blood also expressed DR3 upon activation by IL-33 and engagement of DR3 on activated ILC2 enhanced ILC2 functions. Based on these observations, the authors proposed potential benefit of DR3 engagement in metabolic syndromes including type 2 diabetes mellitus.

Although expression of DR3 on ILC2 and co-stimulatory activity of DR3 engagement has been reported by others in the lung inflammation, the role of DR3 in adipose tissue homeostasis was not explored. This paper contains some new information for readers. There are, however, several points that need to be addressed.

1) The authors used C57BL/6J mouse as a wild type control mice. However, Rag2 knockout (KO) mice, Rag2II2rg double KO mice and Il13 KO mice were on a BALB/c background. Fasting glucose level of HFD-fed DR3 agonist-treated mice was lower than HFD-fed PBS-treated mice in ITT shown in Fig. 2e but such difference was not observed in Rag2 KO or Rag2II2rg double KO mice in Figs. 3f and 4f. Such difference could be due to the difference in genetic backgrounds. The authors should use BALB/c mouse as a wild type control mouse to show data currently shown in Fig. 2.

2) In addition, it is unclear why the authors used C57BL/6J mice instead of C57BL/6N mice for both in vivo and in vitro experiments. It has been widely known that C57BL/6J mice possess a nicotinamide nucleotide transhydrogenase (Nnt) gene mutation and are known to develop obesity and metabolic diseases more readily than C57BL/6N mice can on HFD feeding. Therefore, people usually use C57BL/6N mice for metabolism study.

3) According to the Method section, the authors obtained only C57BL/6J mice as wild type mice. Did the authors transfer ILC2 from C57BL/6J wild type mice into Rag2II2rg double KO mice on a BALB/c background in Fig. 5?

4) Because activation of ILC2 to induce IL-5 and IL-13 contributes to the pathophysiology of allergic inflammation such as asthma, DR3 engagement leading to the secretion of IL-5 and IL-13 could deteriorate certain diseases. The authors should discuss such potential drawback of treatment of T2DM by DR3 engagement.

Following are other points.

1) In Fig. 2f, the authors showed that the mRNA levels were increased upon HFD feeding. Because IL-33 secretion depends on the cellular damages, amounts mRNA do not necessarily correlate with those of secreted proteins. Is there any evidence that secreted IL-33 was also increased by HFD feeding?

2) The headline on the line 154 of page 7 should be modified because the experimental results in this section showed that the effects of DR3 engagement is independent of T/B cells but did not show the dependency on ILC2.

3) In Fig. 5c, was the difference between wild type (red squares) and Il5 KO (purple diamonds) significant?

4) Nagashima et al. (PMID: 29427641) have demonstrated the activating role of GITR in lung

ILC2. The authors should cite this paper in the line 108-109 on page 6 as "Since our group and others recently reported that TNFR2<31> and GITR<30,Nagashima et al.>, the two members".

5) On page 14, references 48-51 should be cited on the line 335 after ILC2s.

6) Stock numbers should be provided for mice obtained from the Jackson Laboratory.

6) In the discussion section on page 14, the authors implied that new flux of naïve ILC2s from the bone marrow. However, parabiosis analyses had shown that new flux contributes little to the number of ILC2 in the adipose tissues and even in the lung after IL-33 administration. It is possible that number of tissue-resident ILC2 in adipose tissues simply decreases under the type 1 environment rather than impaired survival of naïve ILC2 from the bone marrow.

Reviewer #2:

Remarks to the Author:

Activation of Type 2 innate lymphoid cells (ILC2s) has been reported to improve obesity and glucose intolerance, but the mechanisms still remain unclear. The authors found that Death Receptor 3 (DR3) was expressed in ILC2s and induced by IL-33 treatment. DR3 agonist improved insulin resistance and glucose intolerance in high fat diet-fed mice. When the ILC2s of wild-type mice were transferred into the Rag / IL2rg double-deficient mice, DR3 agonist improved insulin resistance and glucose intolerance, but not obesity, in the high-fat fed Rag / IL2rg double-deficient mice. When the ILC2s of IL-13 deficient mice were transferred into the Rag / IL2rg double-deficient mice, these phenotypes such as insulin resistance and glucose intolerance were not improved by DR3 agonist. In human ILC2s, DR3 expression was increased by IL-33.

Major concerns

1. Were the expression of DR3 and its downstream signaling impaired in ILC2s of obesity?

2. As described in the discussion, the number of ILC2s has been reported to be decreased in adipose tissue of obesity. Did the number of ILC2s in adipose tissue change before and after administration of DR3 agonist?

3. Did ILC2-specific DR3 deficient mice exhibit glucose intolerance or insulin resistance?

4. Although DR3 agonist did not change the body weight under high fat-diet condition, did it affect the size and number of fat cells in adipose tissue? Were insulin resistance in liver and skeletal muscle improved by DR3 agonist?

5. ILC2s have been reported to promote beiging of white adipose tissue (Nature 519 :242-6, 2015). Please examine the oxygen consumption and the expression of tyrosine hydroxylase (TH) in adipose tissue when DR3 agonist is injected into the high fat-diet fed mice.

6. By what mechanism does IL-33 increase the DR3 expression?

7. Recently, Sasaki et al. have reported that intestinal ILC2s, but not adipocyte ILC2s, regulates obesity (Cell Rep. 28:202-217.e7, 2019). Please check and discuss DR3 expression in intestinal ILC2s.

Minor concern

1. In page 12 lines 278-279, does "canonical and canonical NF-kB" mean "canonical and uncanonical NF-kB"?

2. In page 10 line 227, "Lastly" is redundant.

Reviewer #3:

Remarks to the Author:

Shafiei-Jahani and colleagues describe a role for DR3 in a model of type 2 diabetes mellitus (T2DM), proposing that stimulation of ILC2s via agonistic anti-DR3 Abs impairs the formation of visceral adipose tissue, ameliorating glucose intolerance and protecting against insulin resistance in animals on a high fat diet and in an IL-13-dependent fashion. DR3 and its capacity to drive IL-5 and IL-13 production from ILC2s is not novel as it has been shown by others in models of other inflammatory diseases, but this is the first demonstration in T2DM. The data shows a very clear phenotype that is potentially very exciting, but there are a number of points, addressing of which would give a better context to their data.

General Points

- (i) The authors need to be a bit more careful with the description of their data. The published functions and expression patterns for DR3 are more extensive than described by the authors in their Introduction (this requires additional references) and this needs coverage and consideration when the authors discuss their data and its potential as a therapeutic avenue. This is not an exhaustive list, but a role for DR3 on stroma in general but also including fibroblasts and epithelial cells specifically, and also myeloid lineage cells (macrophages, neutrophils) and neurones all have been described, so a phenotype following the application of a DR3 agonist to RAG2^{-/-} or RAG2^{-/-}IL2rg^{-/-} mice may not just be due to its impact on ILC2s as described by the authors on lines 179-184 (although the later adoptive transfer expts in Fig 5 are consistent with that effect). Stand alone, there is no ILC2 specific data for the expts described in Figures 2-4. Did the authors look at VAT ILC2s numbers in their DR3 agonist expts in BL/6, RAG2^{-/-} and RAG2^{-/-}IL2rg^{-/-} mice to show accumulation in the VAT of endogenous ILC2s? The ideal expt would be one on ILC2 deficient mice (Rorasg/floxII7rCre/+). Have the authors also considered performing high fat diet expts on DR3^{-/-} mice (which should then have impaired control of glucose)?
- (ii) Relating to the above but perhaps tangential to the main conclusions of the manuscript, there are also aspects of their phenotype which could be caused by the interplay between the cytokine production induced by DR3 activation on ILC2s and non-lymphoid tissue, particularly the reduced numbers of VAT ILC2s when adding IL-5^{-/-} or IL-13^{-/-} ILC2s. The authors comment that this is intriguing, but do not seem to discuss it. Did the authors consider adoptive transfer of DR3^{-/-} ILC2 to give an indication of the underlying mechanisms behind this (ie. to what extent does DR3 signalling in the ILC2s and interactions elsewhere contribute to their accumulation in VAT?).
- (iii) Statistically, t-Tests for individual time points are incorrect if the samples come longitudinally from the same animal as the data suggests. This should be corrected where required.

Specific minor points

- (iv) Line 77 - TL1A is not the only published ligand for DR3 – it is the only known ligand that is a member of the TNF superfamily. Progranulin/attstrin has also been reported to bind DR3 and this requires some coverage/discussion.
- (v) Fig 2F – it would be nice to have protein data for IL-33 as well if possible (rather than just mRNA).
- (vi) Line 227 – one of the 'lastly's should be deleted.
- (vii) Is there any particular reason why the glucose levels in the RAG2^{-/-} mice in Figures 3 and 4 don't seem to match? In the therapeutic expt, they start at twice the level as the initial DR3 engagement expt.
- (viii) Figs 5b-d – the legend does not match the Figures (In the legend the IL-13^{-/-} ILC2 line is green, while lines in the Figs are orange).
- (ix) Fig 7c – current colors make the lines on the histogram difficult to differentiate. This should be altered to make it clearer

We thank the reviewers for their valuable time and thoughtful feedbacks. We appreciate all of the comments that have helped us enhance the context of the results and strengthen the conclusions of this study. Please see our point-by-point response below.

Reviewers' comments:

Reviewer #1 (Remarks to the Author):

1) The authors used C57BL/6J mouse as a wild type control mice. However, Rag2 knockout (KO) mice, Rag2Il2rg double KO mice and Il13 KO mice were on a BALB/c background. Fasting glucose level of mice was lower than HFD-fed PBS-treated mice in ITT shown in Fig. 2e but such difference was not observed in Rag2 KO or Rag2Il2rg double KO mice in Figs. 3f and 4f. Such difference could be due to the difference in genetic backgrounds. The authors should use BALB/c mouse as a wild type control mouse to show data currently shown in Fig. 2.

We agree genetic background in pre-clinical models may influence the severity of diet-induced obesity¹. However, we utilized an established protocol as described by our group and others before²⁻⁶. In **Fig. 2** we used C57BL/6J males, which are known to be susceptible to induction of HFD-induced T2DM. For studies related to *Rag2*^{-/-} and *Rag2*^{-/-} *γc*^{-/-} mice, every strain used is from the BALB/c background. We have now clarified the genetic backgrounds of all mice in the methods section.

We agree with the reviewer that the difference in the fasting glucose levels (between the HFD-Fed Isotype-treated cohort and others) could be due to the genetic profiles of the different backgrounds and experimental variability. It is important to note that although the basal levels may vary, the slopes (response), trends, and overall phenotypes associated with amelioration of T2DM via DR3-dependent stimulation of ILC2s in different cohorts are significant and consistent. To address this concern, we have performed a novel set of experiments using BALB/c mice as suggested and compared the results to our observations presented in **Fig. 2**. The new data is now incorporated to the manuscript as **Supplementary Figure 1**. Briefly, in these experiments a cohort of BALB/c mice were fed either a normal chow diet (NCD) or an HFD for 14 weeks. During this period, the mice were intraperitoneally treated with either DR3 agonist (1mg/mouse) or the isotype control, according to the protocol presented as **Supplementary Figure 1a**. These new results importantly confirm our observations in C57BL/6J as the DR3 agonist-treated cohort showed significant reduction in fasting blood glucose levels when compared to isotype control (**Supplementary Figure 1c**). Furthermore, the glucose tolerance (**Supplementary Figure 1d**) and insulin sensitivity (**Supplementary Figure 1e**) of the DR3 agonist treated BALB/c cohort significantly improved when compared to the isotype control. Altogether, these results suggest that DR3-dependent stimulation of ILC2s significantly improved glucose homeostasis and insulin resistance regardless of the genetic background.

2) In addition, it is unclear why the authors used C57BL/6J mice instead of C57BL/6N mice for both in vivo and in vitro experiments. It has been widely known that C57BL/6J

mice possess an nicotinamide nucleotide transhydrogenase (Nnt) gene mutation and are known to develop obesity and metabolic diseases more readily than C57BL/6N mice can on HFD feeding. Therefore, people usually use C57BL/6N mice for metabolism study.

We appreciate the reviewer's comment regarding the differences between the sub-strains of C57BL/6. In this study, we followed an established model of C57BL/6J that has been previously published by our group and several others²⁻⁶. We agree the M35T missense mutation alters RNA splicing of *Nnt* gene and allows for a more rapid development of metabolic distress^{7,8}. Therefore, we believe C57BL/6J and C57BL/6N should not be used interchangeably within the same study. We have now clarified the genetic backgrounds of all mice in the manuscript and included the therapeutic efficacy of DR3 engagement in two independent genetic backgrounds (BALB/c and C57BL/6J).

3) According to the Method section, the authors obtained only C57BL/6J mice as wild type mice. Did the authors transfer ILC2 from C57BL/6J wild type mice into Rag2Il2rg double KO mice on a BALB/c background in Fig. 5?

We apologize for not being clear and have now clarified the genetic background of donors and recipients in the methods section. WT, *Il5*^{-/-}, *Il13*^{-/-} donors and *Rag2*^{-/-} *yc*^{-/-} recipients were all on BALB/c background as described in the methods.

4) Because activation of ILC2 to induce IL-5 and IL-13 contributes to the pathophysiology of allergic inflammation such as asthma, DR3 engagement leading to the secretion of IL-5 and IL-13 could deteriorate certain diseases. The authors should discuss such potential drawback of treatment of T2DM by DR3 engagement.

This is a great point raised by the reviewer and we have now discussed this potential adverse effect in the manuscript. We believe local administration and hydrophobic modifications need to be developed and explored to minimize any adverse effects associated with DR3 treatment for T2DM.

1) In Fig. 2f, the authors showed that the mRNA levels were increased upon HFD feeding. Because IL-33 secretion depends on the cellular damages, amounts mRNA do not necessarily correlate with those of secreted proteins. Is there any evidence that secreted IL-33 was also increased by HFD feeding?

Unfortunately, we made numerous attempts to detect local IL-33 at protein level but it appears that the levels are below the threshold limits of ELISA and Luminex. Therefore, we designed RT-PCR methods and observed a significant increase of IL-33 at transcriptome level (**Fig. 2f**).

2) The headline on the line 154 of page 7 should be modified because the experimental results in this section showed that the effects of DR3 engagement is independent of T/B cells but did not show the dependency on ILC2.

We have now modified and improved the headline.

3) In Fig. 5c, was the difference between wild type (red squares) and *I15* KO (purple diamonds) significant?

The trend observed between wild type ILC2 and *I15*^{-/-} ILC2 receipt cohorts did not reach statistical significance. We now added ns (not significant) to the Figure for clarification.

4) Nagashima et al. (PMID: 29427641) have demonstrated the activating role of GITR in lung ILC2. The authors should cite this paper in the line 108-109 on page 6 as “Since our group and others recently reported that TNFR2<31> and GITR<30>, Nagashima et al.>, the two members”.

We have now cited the study by Nagashima *et al.* in the manuscript.

5) On page 14, references 48-51 should be cited on the line 335 after ILC2s.

References 48-51 are now cited as recommended.

6) Stock numbers should be provided for mice obtained from the Jackson Laboratory.

We have included the stock numbers in the methods.

7) In the discussion section on page 14, the authors implied that new flux of naïve ILC2s from the bone marrow. However, parabiosis analyses had shown that new flux contributes little to the number of ILC2 in the adipose tissues and even in the lung after IL-33 administration. It is possible that number of tissue-resident ILC2 in adipose tissues simply decreases under the type 1 environment rather than impaired survival of naïve ILC2 from the bone marrow.

This is an interesting topic of discussion. Trafficking of ILC2s remains under investigation and is an area of active research. We are aware of parabiosis experiments demonstrating that tissue resident ILC2s are programmed and destined for specific tissues. However, the results of those experiments clearly suggest that bone marrow supply is necessary for homeostatic maintenance of tissue resident ILC2s. Please note that although tissue-residents ILC2s do not traffic to other tissues, there has been several lines of work that show BM derived ILC2s have the ability to traffic and replenish ILC2s in various tissues¹⁵⁻¹⁷. We now discuss this point in the manuscript.

We agree with the reviewer that type 1 environment is detrimental to survival of ILC2s and contributes to the low number and insufficient activation of ILC2s in VAT. However, we also believe that BM ILC2 flux is needed to adequately sustain homeostasis of tissue resident ILC2s. We have now clarified this point in the manuscript.

Reviewer #2 (Remarks to the Author):

Major concerns

1. Were the expression of DR3 and its downstream signaling impaired in ILC2s of obesity?

This is a great question. We have assessed and compared DR3 expression in naïve and activated ILC2s of mice (HFD and NCD) and incorporated the results as **Figures 1a-c**. Our data suggest that DR3 expression is comparable on VAT ILC2s. Moreover, we also assessed the expression levels of downstream transcription factors p65 and p52 NF- κ B pathways and compared the levels in ILC2s derived from adipose tissue of HFD and NCD groups (**Supplementary Figure 4**). Our data suggest that DR3 dependent NF- κ B signaling pathways are intact and also comparable in both groups.

2. As described in the discussion, the number of ILC2s has been reported to be decreased in adipose tissue of obesity. Did the number of ILC2s in adipose tissue change before and after administration of DR3 agonist?

We quantified the number of ILC2s in the WT and *Rag2*^{-/-} mice treated with DR3 agonist or isotype control. We added the quantification results for WT mice as **Fig. 2g** and **Supplementary Figure 2a** and for *Rag2*^{-/-} mice as **Supplementary Figure 2b-c**. While we observed that the ILC2 numbers were lower in obese group, DR3 treatment significantly and consistently increased the number of adipose ILC2s in both WT and *Rag2*^{-/-} mice treated with DR3 agonist.

3. Did ILC2-specific DR3 deficient mice exhibit glucose intolerance or insulin resistance?

We considered to explore *DR3*^{-/-} ILC2s at the beginning of this project. However, it became apparent that *DR3*^{-/-} mice have a very low number of ILC2 as DR3 plays an important role in ILC2 homeostasis and cytokine production^{18,19}. Therefore, isolation of adequate number of VAT ILC2s from *DR3*^{-/-} mice was not technically feasible for performing adoptive transfer experiments. We need to point out that our results highlight the benefit and phenotype of DR3 engagement and signaling on ILC2s. The results of experiments with DR3 deficient mice would only provide information regarding the specificity of murine DR3 agonist, which was addressed by other groups before²⁰⁻²³.

4. Although DR3 agonist did not change the body weight under high fat-diet condition, did it affect the size and number of fat cells in adipose tissue? Were insulin resistance in liver and skeletal muscle improved by DR3 agonist?

We thank the reviewer for their comment. We performed additional histological analysis of VAT in both preventive (**Fig. 3j-l**) and therapeutic models (**Fig. 4. g-i**). Consistent with the improved insulin resistance and glucose tolerance previously shown in our study, the DR3-treated cohort in both models exhibited reduced hypertrophy of adipocytes and

abrogated adipose tissue hyperplasia. Since hypertrophy and hyperplasia of adipocytes have been implicated in obese patients²⁴⁻²⁶, these new results add yet another supporting evidence for the efficacy and exciting therapeutic potential of DR3 agonistic treatment.

Regarding the diversity of mechanisms by which activation of ILC2s ameliorates phenotypes associated with T2DM, several groups have already demonstrated that activation of ILC2s mainly ameliorates T2DM by promoting beiging of white adipose tissue through production of Th2 associated cytokines^{5,6,27}. We closely checked the liver mass in our experimental *Rag2*^{-/-} groups and did not detect a significant difference in hepatic mass. This result is now added as **Supplementary Figure 2d**. Unfortunately, we do not have the data or samples available to assess the skeletal muscle and lean mass. However, we recently showed that activation of ILC2s via GITR, another member of the TNF superfamily increased the lean mass of the mice and ameliorated T2DM⁶. We now discuss this possibility and cite the relevant literature in the manuscript. Additionally, we have added new data to further examine the mechanisms of beiging and increased metabolic rate in response to the reviewer's comments below.

5. ILC2s have been reported to promote beiging of white adipose tissue (Nature 519 :242-6, 2015). Please examine the oxygen consumption and the expression of tyrosine hydroxylase (TH) in adipose tissue when DR3 agonist is injected into the high fat-diet fed mice.

We believe our results are in agreement with Brestoff's report that ILC2 activation can induce beiging of the adipose tissue and in turn increase thermogenesis and caloric expenditure. In this study, we previously showed the expression of uncoupling protein 1 (a characteristic gene of brown adipose tissue) and presence of VAT-associated alternatively activated macrophages (AAMs) are augmented, suggesting that DR3 treatment increases beiging of white adipose tissue (**Fig. 3g-h**). To further strengthen our observations, we performed additional RT-qPCR experiments and have now quantified five additional thermogenic genes in the VAT. These five genes (*Cidea*, *Prdm16*, *Pgc1a*, *Cox7a*, *Dio2*) have been shown to be molecular markers of brown adipose tissue. Moreover, *Dio2*, *Prdm16* and *Pgc1a* expressions are also positively associated with a higher metabolic rate²⁸⁻³⁰. Our results suggest that the expression of aforementioned alternatively activated macrophages and genes in VAT lysate was significantly increased after DR3 treatment. These results are now added as **Fig. 3i** to the manuscript.

Next, we assessed the expression of tyrosine hydroxylase (*TH*) and 5 well-known genes encoding respiratory chain complexes (I, III, IV, and V) in the adipose tissue of the mice treated with DR3 agonist or isotype. We chose this approach due to logistical constraints. These new RT-qPCR results demonstrate that DR3 agonist treatment significantly increased tyrosine hydroxylase (*TH*); as well as complex I (*ND1* and *ND5*), complex III (mitochondria-encoded NADH dehydrogenase I (*Cytb*)), complex IV (mitochondria-encoded cytochrome c oxidase I (*Cox1*)), and complex V (mitochondria-encoded ATP synthase 6 (*Atp6*)) in the adipose tissue. These results are now added to

the manuscript as **Supplementary Figure 2e-j**. Taken together, these results are in agreement with the findings by Brestoff *et al.* and support the notion that activation of ILC2s induces beiging of the adipose tissue and increase the metabolic rate of the VAT.

6. By what mechanism does IL-33 increase the DR3 expression?

IL-33 is a pluripotent cytokine that is capable of activating multiple cellular signaling networks³¹. Our results suggest that DR3 engagement can act on ILC2s independent of IL-33 signaling, as naive ILC2s in both mice and humans showed higher cytokine production after treatment with DR3 agonist or TL1A-L (**Fig 1d.** and **Fig 7d.**). However, IL-33 upregulates a variety of pathways including NF- κ B signaling via MyD88/IRAK/TRAF6, ERK1/2, JNK, p38 and PI3K/AKT, and all these pathways have been previously shown to upregulate costimulatory molecules including (but not limited to) CD40, CD80, CD86, OX40, ICAM-1, PD-1, TNFR2 and now DR3³²⁻⁴⁵. We now discuss the effect of IL-33 on enhancement of costimulatory molecules in the discussion and cite relevant articles.

7. Recently, Sasaki *et al.* have reported that intestinal ILC2s, but not adipocyte ILC2s, regulates obesity (Cell Rep. 28:202-217.e7, 2019). Please check and discuss DR3 expression in intestinal ILC2s.

The expression of DR3 on intestinal resident ILCs was previously reported and data suggest that DR3 expression is comparable to the VAT resident ILC2s. We believe tissue-specific programming and DR3-dependent stimulation need to be investigated in another study. However, we agree with the reviewer and we now discuss tissue-specific programming of ILC2s, and the studies conducted by Sasaki *et al.* and Li *et al.* in the manuscript.

Minor concern

1. In page 12 lines 278-279, does “canonical and canonical NF- κ B” mean “canonical and uncanonical NF- κ B”?

We have revised the manuscript.

2. In page 10 line 227, “Lastly” is redundant.

We have adapted the manuscript accordingly.

Reviewer #3 (Remarks to the Author):

General Points

(i) The authors need to be a bit more careful with the description of their data. The published functions and expression patterns for DR3 are more extensive than described

by the authors in their Introduction (this requires additional references) and this needs coverage and consideration when the authors discuss their data and its potential as a therapeutic avenue. This is not an exhaustive list, but a role for DR3 on stroma in general but also including fibroblasts and epithelial cells specifically, and also myeloid lineage cells (macrophages, neutrophils) and neurones all have been described, so a phenotype following the application of a DR3 agonist to RAG2^{-/-} or RAG2^{-/-}IL2rg^{-/-} mice may not just be due to its impact on ILC2s as described by the authors on lines 179-184 (although the later adoptive transfer expts in Fig 5 are consistent with that effect). Stand alone, there is no ILC2 specific data for the expts described in Figures 2-4. Did the authors look at VAT ILC2s numbers in their DR3 agonist expts in BL/6, RAG2^{-/-} and RAG2^{-/-}IL2rg^{-/-} mice to show accumulation in the VAT of endogenous ILC2s? The ideal expt would be one on ILC2 deficient mice (Rorasg/flox117rCre/+). Have the authors also considered performing high fat diet expts on DR3^{-/-} mice (which should then have impaired control of glucose)?

We thank the reviewer for the thoughtful comments. We agree that the expression landscape of DR3 is beyond ILC2s. We now discussed this important point and cited appropriate studies demonstrating the expression of DR3 on neurons, fibroblasts, epithelial cells, and myeloid lineage cells^{18,19,51-62} (Please see introduction paragraph 2).

The reviewer 3's inquiry regarding the number of ILC2s was addressed in response to reviewer 2's second comment above. Briefly, we have added new data that suggest DR3 stimulation of new flux of naïve ILC2s and co-stimulation of endothelial-derived IL-33-activated VAT ILC2s (through the canonical and/or non-canonical NF-κB pathways) efficiently combats the decline in number of VAT ILC2s reported in animal models and obese patients^{5,63}. As additionally mentioned above, we now added new data that demonstrate DR3 expression and signaling is not impaired in the HFD model, further underscoring the therapeutic viability of targeting this pathway in obese patients.

Unfortunately, we do not have access to *Rorasg/flox117rCre/+*, which would have been an excellent tool for confirming previous studies that have showed importance of ILC2s in T2DM^{5,27,64}. The reviewer 3's inquiry regarding *DR3^{-/-}* mice was addressed in our response to reviewer 2's third comment above. As discussed above *DR3^{-/-}* mice have extremely low numbers of ILC2s as reported by several groups^{18,19}. Therefore, it is not technically feasible to isolate sufficient numbers of VAT *DR3^{-/-}* ILC2s to perform experiments. Nevertheless, the results of adoptive transfer with *DR3^{-/-}* ILC2s treated with DR3 agonist or isotype control will only provide information regarding the specificity of the murine DR3 agonist, which was described by several other groups before²⁰⁻²³. We need to point out that our results highlight the benefit and phenotype of DR3 engagement on ILC2s and provide a potentially therapeutic approach for the patients with T2DM.

(ii) Relating to the above but perhaps tangential to the main conclusions of the manuscript, there are also aspects of their phenotype which could be caused by the interplay between the cytokine production induced by DR3 activation on ILC2s and non-

lymphoid tissue, particularly the reduced numbers of VAT ILC2s when adding IL-5^{-/-} or IL-13^{-/-} ILC2s. The authors comment that this is intriguing, but do not seem to discuss it. Did the authors consider adoptive transfer of DR3^{-/-} ILC2 to give an indication of the underlying mechanisms behind this (ie. to what extent does DR3 signalling in the ILC2s and interactions elsewhere contribute to their accumulation in VAT?).

We agree with the reviewer that the interplay between ILC2s and non-lymphoid tissue is important and needs to be discussed. Recent studies have highlighted the critical interplays between ILC2s and their respective physical tissue niches during homeostasis and inflammation^{49,65-68}. We agree with the reviewer that cytokines produced by ILC2s (particularly, IL-5 and IL-13) will impact somatic cells such as adipocytes. A study by Dahlgren *et al.* demonstrated adventitial stromal cells (ASCs) derived IL-33 and TSLP regulate ILC2 effector function and expansion⁶⁵. Additionally, the interactions between stroma and adipose-resident leukocytes may be mediated by adhesion molecules. Recently, Rana *et al.* showed that white adipose tissue-resident multipotent stromal cells (WAT-MSCs) supports ICAM-1-mediated proliferation and activation of ILC2s⁴⁹. We now discuss the interplay between ILC2s and the local stromal cell niche in the manuscript and cite the relevant studies.

As discussed above, one of the resulting consequences of activating ILC2 effector function is the conversion of white to beige adipose tissue. As previously described, beiging of the adipose tissue increases thermogenesis, and subsequently increases the caloric expenditure⁵. As mentioned in response to reviewer 2, we have added the data for five additional thermogenic markers (*Cidea*, *Prdm16*, *Pgc1a*, *Cox7a*, *Dio2*) and 5 known genes encoding respiratory chain complexes (I, III, IV, and V) in VAT. These data are now added as **Fig. 3i** and **Supplementary Figure 2e-j**. Since the adoptive transfer experiments in **Fig. 5** indicated that DR3 based amelioration of T2DM is IL-13⁺ ILC2s dependent, our results collectively indicate DR3 agonistic treatment improves insulin sensitivity and glucose tolerance by promoting beiging of the adipose tissue and increasing the metabolic rate of the VAT. Moreover, we have now added new data showing reduced adipose hypertrophy and hyperplasia, further indicating DR3 treatment is effective in both preventative and therapeutic *Rag2^{-/-}* models (**Fig. 3j-l** and **Fig4. g-i**). Taken all together, we believe DR3 agonistic treatment results in a potent and efficient therapeutic strategy to prevent and reverse the metabolic syndromes associated with T2DM.

(iii) Statistically, t-Tests for individual time points are incorrect if the samples come longitudinally from the same animal as the data suggests. This should be corrected where required.

We thank the reviewer for the thoughtful comment. After discussion with biostatisticians at USC, we have now implemented t-Tests, one-way and two-way ANOVA tests as appropriate for various results based on the nature of experiments. We have updated the manuscript by adding the new statistical results.

Specific minor points

(iv) Line 77 - TL1A is not the only published ligand for DR3 – it is the only known ligand that is a member of the TNF superfamily. Progranulin/attstrin has also been reported to bind DR3 and this requires some coverage/discussion.

We thank the reviewer for raising this point, and we have now updated the manuscript to address the studies related to DR3 and progranulin-derived Atsttrin.

(v) Fig 2F – it would be nice to have protein data for IL-33 as well if possible (rather than just mRNA).

As mentioned in response to reviewer 1's first minor point above, we made several attempts to detect IL-33 from adipocyte lysate but unfortunately it has become apparent that the tissue levels of cytokines such as IL-33 are very low and below the threshold limits of detection by ELISA or Luminex.

(vi) Line 227 – one of the 'lastly's should be deleted.

We edited line 227.

(vii) Is there any particular reason why the glucose levels in the RAG2^{-/-} mice in Figures 3 and 4 don't seem to match? In the therapeutic expt, they start at twice the level as the initial DR3 engagement expt.

As mentioned in the manuscript, each experiment is a representative of 3 independent studies that were carried out at different times. Although the slope, response and major phenotypic response to treatment has been the same in each of the replicate studies, the glucose levels appear to be variable among the replicates. We encountered similar basal variations in our previous studies⁶. Although not yet comprehensibly understood, the glucose levels appear mouse batch dependent. To the best of our knowledge, it remains to be unraveled whether this variation is due to background genetic drift or environmental variables. We agree this could lead to interesting future studies, but we believe exploration of the diverse set of factors that contribute to base line glucose variation is out of the scope of the current study. Since DR3 treatment effectively improved the overall metabolic phenotype in all of the different experimental setups, any factors contributing to the mentioned variation are likely independent of DR3 signaling on ILC2s.

(viii) Figs 5b-d – the legend does not match the Figures (In the legend the IL-13^{-/-} ILC2 line is green, while lines in the Figs are orange).

We thank the reviewer for the comment and have now changed the colors in **Fig. 5**.

(ix) Fig 7c – current **colors** make the lines on the histogram difficult to differentiate. This should be altered to make it clearer

We altered the color scheme in **Fig. 7c** as recommended by the reviewer.

REFERENCES

- 1 Fisher-Wellman, K. H. *et al.* A Direct Comparison of Metabolic Responses to High-Fat Diet in C57BL/6J and C57BL/6NJ Mice. *Diabetes* **65**, 3249-3261, doi:10.2337/db16-0291 (2016).
- 2 Surwit, R. S., Kuhn, C. M., Cochrane, C., McCubbin, J. A. & Feinglos, M. N. Diet-induced type II diabetes in C57BL/6J mice. *Diabetes* **37**, 1163-1167, doi:10.2337/diab.37.9.1163 (1988).
- 3 Gallou-Kabani, C. *et al.* C57BL/6J and A/J mice fed a high-fat diet delineate components of metabolic syndrome. *Obesity (Silver Spring)* **15**, 1996-2005, doi:10.1038/oby.2007.238 (2007).
- 4 Yang, Y., Smith, D. L., Jr., Keating, K. D., Allison, D. B. & Nagy, T. R. Variations in body weight, food intake and body composition after long-term high-fat diet feeding in C57BL/6J mice. *Obesity (Silver Spring)* **22**, 2147-2155, doi:10.1002/oby.20811 (2014).
- 5 Brestoff, J. R. *et al.* Group 2 innate lymphoid cells promote beiging of white adipose tissue and limit obesity. *Nature* **519**, 242-246, doi:10.1038/nature14115 (2015).
- 6 Galle-Treger, L. *et al.* Costimulation of type-2 innate lymphoid cells by GITR promotes effector function and ameliorates type 2 diabetes. *Nature communications* **10**, 713, doi:10.1038/s41467-019-08449-x (2019).
- 7 Freeman, H. C., Hugill, A., Dear, N. T., Ashcroft, F. M. & Cox, R. D. Deletion of nicotinamide nucleotide transhydrogenase: a new quantitative trait locus accounting for glucose intolerance in C57BL/6J mice. *Diabetes* **55**, 2153-2156, doi:10.2337/db06-0358 (2006).
- 8 Huang, T. T. *et al.* Genetic modifiers of the phenotype of mice deficient in mitochondrial superoxide dismutase. *Hum Mol Genet* **15**, 1187-1194, doi:10.1093/hmg/ddl034 (2006).
- 9 Shimomura, K. *et al.* Insulin secretion from beta-cells is affected by deletion of nicotinamide nucleotide transhydrogenase. *Methods Enzymol* **457**, 451-480, doi:10.1016/S0076-6879(09)05025-3 (2009).
- 10 Hurrell, B. P., Shafiei Jahani, P. & Akbari, O. Social Networking of Group Two Innate Lymphoid Cells in Allergy and Asthma. *Front Immunol* **9**, 2694, doi:10.3389/fimmu.2018.02694 (2018).
- 11 Maazi, H. & Akbari, O. Type two innate lymphoid cells: the Janus cells in health and disease. *Immunol Rev* **278**, 192-206, doi:10.1111/imr.12554 (2017).
- 12 Pelaia, C. *et al.* Interleukin-5 in the Pathophysiology of Severe Asthma. *Front Physiol* **10**, 1514, doi:10.3389/fphys.2019.01514 (2019).
- 13 Marone, G. *et al.* The Intriguing Role of Interleukin 13 in the Pathophysiology of Asthma. *Front Pharmacol* **10**, 1387, doi:10.3389/fphar.2019.01387 (2019).

- 14 Brannan, J. D. & Lougheed, M. D. Airway hyperresponsiveness in asthma: mechanisms, clinical significance, and treatment. *Front Physiol* **3**, 460, doi:10.3389/fphys.2012.00460 (2012).
- 15 Gasteiger, G., Fan, X., Dikiy, S., Lee, S. Y. & Rudensky, A. Y. Tissue residency of innate lymphoid cells in lymphoid and nonlymphoid organs. *Science (New York, N.Y.)* **350**, 981-985, doi:10.1126/science.aac9593 (2015).
- 16 Walker, J. A. & McKenzie, A. N. Development and function of group 2 innate lymphoid cells. *Current opinion in immunology* **25**, 148-155, doi:10.1016/j.coi.2013.02.010 (2013).
- 17 Ricardo-Gonzalez, R. R. *et al.* Tissue-specific pathways extrude activated ILC2s to disseminate type 2 immunity. *The Journal of experimental medicine* **217**, doi:10.1084/jem.20191172 (2020).
- 18 Yu, X. *et al.* TNF superfamily member TL1A elicits type 2 innate lymphoid cells at mucosal barriers. *Mucosal Immunol* **7**, 730-740, doi:10.1038/mi.2013.92 (2014).
- 19 Meylan, F. *et al.* The TNF-family cytokine TL1A promotes allergic immunopathology through group 2 innate lymphoid cells. *Mucosal Immunol* **7**, 958-968, doi:10.1038/mi.2013.114 (2014).
- 20 Longman, R. S. *et al.* CX(3)CR1(+) mononuclear phagocytes support colitis-associated innate lymphoid cell production of IL-22. *The Journal of experimental medicine* **211**, 1571-1583, doi:10.1084/jem.20140678 (2014).
- 21 Wolf, D. *et al.* Tregs expanded in vivo by TNFRSF25 agonists promote cardiac allograft survival. *Transplantation* **94**, 569-574, doi:10.1097/TP.0b013e318264d3ef (2012).
- 22 Schreiber, T. H., Wolf, D., Boder, M., Gonzalez, L. & Podack, E. R. T cell costimulation by TNFR superfamily (TNFRSF)4 and TNFRSF25 in the context of vaccination. *Journal of immunology (Baltimore, Md. : 1950)* **189**, 3311-3318, doi:10.4049/jimmunol.1200597 (2012).
- 23 Fang, L., Adkins, B., Deyev, V. & Podack, E. R. Essential role of TNF receptor superfamily 25 (TNFRSF25) in the development of allergic lung inflammation. *The Journal of experimental medicine* **205**, 1037-1048, doi:10.1084/jem.20072528 (2008).
- 24 Verboven, K. *et al.* Abdominal subcutaneous and visceral adipocyte size, lipolysis and inflammation relate to insulin resistance in male obese humans. *Sci Rep* **8**, 4677, doi:10.1038/s41598-018-22962-x (2018).
- 25 Henninger, A. M., Eliasson, B., Jenndahl, L. E. & Hammarstedt, A. Adipocyte hypertrophy, inflammation and fibrosis characterize subcutaneous adipose tissue of healthy, non-obese subjects predisposed to type 2 diabetes. *PLoS One* **9**, e105262, doi:10.1371/journal.pone.0105262 (2014).
- 26 Muir, L. A. *et al.* Adipose tissue fibrosis, hypertrophy, and hyperplasia: Correlations with diabetes in human obesity. *Obesity (Silver Spring)* **24**, 597-605, doi:10.1002/oby.21377 (2016).
- 27 Lee, M. W. *et al.* Activated type 2 innate lymphoid cells regulate beige fat biogenesis. *Cell* **160**, 74-87, doi:10.1016/j.cell.2014.12.011 (2015).
- 28 Marsili, A. *et al.* Mice with a targeted deletion of the type 2 deiodinase are insulin resistant and susceptible to diet induced obesity. *PLoS One* **6**, e20832, doi:10.1371/journal.pone.0020832 (2011).

- 29 Cohen, P. *et al.* Ablation of PRDM16 and beige adipose causes metabolic dysfunction and a subcutaneous to visceral fat switch. *Cell* **156**, 304-316, doi:10.1016/j.cell.2013.12.021 (2014).
- 30 Bostrom, P. *et al.* A PGC1-alpha-dependent myokine that drives brown-fat-like development of white fat and thermogenesis. *Nature* **481**, 463-468, doi:10.1038/nature10777 (2012).
- 31 Pinto, S. M. *et al.* A network map of IL-33 signaling pathway. *J Cell Commun Signal* **12**, 615-624, doi:10.1007/s12079-018-0464-4 (2018).
- 32 Drube, S. *et al.* MK2/3 Are Pivotal for IL-33-Induced and Mast Cell-Dependent Leukocyte Recruitment and the Resulting Skin Inflammation. *Journal of immunology (Baltimore, Md. : 1950)* **197**, 3662-3668, doi:10.4049/jimmunol.1600658 (2016).
- 33 Saluja, R., Hawro, T., Eberle, J., Church, M. K. & Maurer, M. Interleukin-33 promotes the proliferation of mouse mast cells through ST2/MyD88 and p38 MAPK-dependent and Kit-independent pathways. *J Biol Regul Homeost Agents* **28**, 575-585 (2014).
- 34 Funakoshi-Tago, M., Tago, K., Sato, Y., Tominaga, S. & Kasahara, T. JAK2 is an important signal transducer in IL-33-induced NF-kappaB activation. *Cell Signal* **23**, 363-370, doi:10.1016/j.cellsig.2010.10.006 (2011).
- 35 Mun, S. H. *et al.* Interleukin-33 stimulates formation of functional osteoclasts from human CD14(+) monocytes. *Cellular and molecular life sciences : CMLS* **67**, 3883-3892, doi:10.1007/s00018-010-0410-y (2010).
- 36 Drube, S. *et al.* The receptor tyrosine kinase c-Kit controls IL-33 receptor signaling in mast cells. *Blood* **115**, 3899-3906, doi:10.1182/blood-2009-10-247411 (2010).
- 37 Lingel, A. *et al.* Structure of IL-33 and its interaction with the ST2 and IL-1RAcP receptors--insight into heterotrimeric IL-1 signaling complexes. *Structure* **17**, 1398-1410, doi:10.1016/j.str.2009.08.009 (2009).
- 38 Choi, Y. S. *et al.* Interleukin-33 induces angiogenesis and vascular permeability through ST2/TRAF6-mediated endothelial nitric oxide production. *Blood* **114**, 3117-3126, doi:10.1182/blood-2009-02-203372 (2009).
- 39 Funakoshi-Tago, M. *et al.* TRAF6 is a critical signal transducer in IL-33 signaling pathway. *Cell Signal* **20**, 1679-1686, doi:10.1016/j.cellsig.2008.05.013 (2008).
- 40 Ashlin, T. G. *et al.* The anti-atherogenic cytokine interleukin-33 inhibits the expression of a disintegrin and metalloproteinase with thrombospondin motifs-1, -4 and -5 in human macrophages: Requirement of extracellular signal-regulated kinase, c-Jun N-terminal kinase and phosphoinositide 3-kinase signaling pathways. *Int J Biochem Cell Biol* **46**, 113-123, doi:10.1016/j.biocel.2013.11.008 (2014).
- 41 Hurrell, B. P. *et al.* TNFR2 Signaling Enhances ILC2 Survival, Function, and Induction of Airway Hyperreactivity. *Cell Rep* **29**, 4509-4524 e4505, doi:10.1016/j.celrep.2019.11.102 (2019).
- 42 Taylor, S. *et al.* PD-1 regulates KLRG1(+) group 2 innate lymphoid cells. *The Journal of experimental medicine* **214**, 1663-1678, doi:10.1084/jem.20161653 (2017).
- 43 Molofsky, A. B., Savage, A. K. & Locksley, R. M. Interleukin-33 in Tissue Homeostasis, Injury, and Inflammation. *Immunity* **42**, 1005-1019, doi:10.1016/j.immuni.2015.06.006 (2015).

- 44 Dominguez, D. *et al.* Exogenous IL-33 Restores Dendritic Cell Activation and Maturation in Established Cancer. *Journal of immunology (Baltimore, Md. : 1950)* **198**, 1365-1375, doi:10.4049/jimmunol.1501399 (2017).
- 45 Wallrapp, A. *et al.* The neuropeptide NMU amplifies ILC2-driven allergic lung inflammation. *Nature* **549**, 351-356, doi:10.1038/nature24029 (2017).
- 46 Li, J. *et al.* Activation of DR3 signaling causes loss of ILC3s and exacerbates intestinal inflammation. *Nature communications* **10**, 3371, doi:10.1038/s41467-019-11304-8 (2019).
- 47 Ricardo-Gonzalez, R. R. *et al.* Tissue signals imprint ILC2 identity with anticipatory function. *Nature immunology* **19**, 1093-1099, doi:10.1038/s41590-018-0201-4 (2018).
- 48 Yudanin, N. A. *et al.* Spatial and Temporal Mapping of Human Innate Lymphoid Cells Reveals Elements of Tissue Specificity. *Immunity* **50**, 505-519 e504, doi:10.1016/j.immuni.2019.01.012 (2019).
- 49 Rana, B. M. J. *et al.* A stromal cell niche sustains ILC2-mediated type-2 conditioning in adipose tissue. *The Journal of experimental medicine* **216**, 1999-2009, doi:10.1084/jem.20190689 (2019).
- 50 Sasaki, T. *et al.* Innate Lymphoid Cells in the Induction of Obesity. *Cell Rep* **28**, 202-217 e207, doi:10.1016/j.celrep.2019.06.016 (2019).
- 51 McLaren, J. E. *et al.* The TNF-like protein 1A-death receptor 3 pathway promotes macrophage foam cell formation in vitro. *Journal of immunology (Baltimore, Md. : 1950)* **184**, 5827-5834, doi:10.4049/jimmunol.0903782 (2010).
- 52 Meylan, F. *et al.* The TNF-family receptor DR3 is essential for diverse T cell-mediated inflammatory diseases. *Immunity* **29**, 79-89, doi:10.1016/j.immuni.2008.04.021 (2008).
- 53 Twohig, J. P. *et al.* The death receptor 3/TL1A pathway is essential for efficient development of antiviral CD4(+) and CD8(+) T-cell immunity. *FASEB J* **26**, 3575-3586, doi:10.1096/fj.11-200618 (2012).
- 54 Nishikii, H. *et al.* DR3 signaling modulates the function of Foxp3+ regulatory T cells and the severity of acute graft-versus-host disease. *Blood* **128**, 2846-2858, doi:10.1182/blood-2016-06-723783 (2016).
- 55 Heidemann, S. C. *et al.* TL1A selectively enhances IL-12/IL-18-induced NK cell cytotoxicity against NK-resistant tumor targets. *J Clin Immunol* **30**, 531-538, doi:10.1007/s10875-010-9382-9 (2010).
- 56 Ahn, Y. O. *et al.* Human group3 innate lymphoid cells express DR3 and respond to TL1A with enhanced IL-22 production and IL-2-dependent proliferation. *European journal of immunology* **45**, 2335-2342, doi:10.1002/eji.201445213 (2015).
- 57 Twohig, J. P. *et al.* Age-dependent maintenance of motor control and corticostriatal innervation by death receptor 3. *J Neurosci* **30**, 3782-3792, doi:10.1523/JNEUROSCI.1928-09.2010 (2010).
- 58 Jacob, N. *et al.* Inflammation-independent TL1A-mediated intestinal fibrosis is dependent on the gut microbiome. *Mucosal Immunol* **11**, 1466-1476, doi:10.1038/s41385-018-0055-y (2018).
- 59 Al-Lamki, R. S. *et al.* TL1A both promotes and protects from renal inflammation and injury. *J Am Soc Nephrol* **19**, 953-960, doi:10.1681/ASN.2007060706 (2008).

- 60 Al-Lamki, R. S. *et al.* DR3 signaling protects against cisplatin nephrotoxicity mediated by tumor necrosis factor. *Am J Pathol* **180**, 1454-1464, doi:10.1016/j.ajpath.2012.01.003 (2012).
- 61 Wang, E. C. On death receptor 3 and its ligands. *Immunology* **137**, 114-116, doi:10.1111/j.1365-2567.2012.03606.x (2012).
- 62 Valatas, V., Kolios, G. & Bamias, G. TL1A (TNFSF15) and DR3 (TNFRSF25): A Co-stimulatory System of Cytokines With Diverse Functions in Gut Mucosal Immunity. *Front Immunol* **10**, 583, doi:10.3389/fimmu.2019.00583 (2019).
- 63 Molofsky, A. B. *et al.* Interleukin-33 and Interferon-gamma Counter-Regulate Group 2 Innate Lymphoid Cell Activation during Immune Perturbation. *Immunity* **43**, 161-174, doi:10.1016/j.immuni.2015.05.019 (2015).
- 64 Cautivo, K. M. & Molofsky, A. B. Regulation of metabolic health and adipose tissue function by group 2 innate lymphoid cells. *European journal of immunology* **46**, 1315-1325, doi:10.1002/eji.201545562 (2016).
- 65 Dahlgren, M. W. *et al.* Adventitial Stromal Cells Define Group 2 Innate Lymphoid Cell Tissue Niches. *Immunity* **50**, 707-722 e706, doi:10.1016/j.immuni.2019.02.002 (2019).
- 66 Molofsky, A. B. *et al.* Innate lymphoid type 2 cells sustain visceral adipose tissue eosinophils and alternatively activated macrophages. *The Journal of experimental medicine* **210**, 535-549, doi:10.1084/jem.20121964 (2013).
- 67 Han, X. *et al.* Mapping the Mouse Cell Atlas by Microwell-Seq. *Cell* **172**, 1091-1107 e1017, doi:10.1016/j.cell.2018.02.001 (2018).
- 68 Mahlakoiv, T. *et al.* Stromal cells maintain immune cell homeostasis in adipose tissue via production of interleukin-33. *Sci Immunol* **4**, doi:10.1126/sciimmunol.aax0416 (2019).

Reviewers' Comments:

Reviewer #1:

Remarks to the Author:

Shafiei-Jahani et al. performed additional experiments and revised their paper, which was significantly improved. There are, however, several minor points that need to be addressed as listed below.

) Lines 28 and 58: ILC2 are "group" 2 innate lymphoid cells but not "type" 2 innate lymphoid cells (PMID: 23348417, 30142344).

2) Line 189: Ref. 29 is not related to insulin-resistance but to asthma. The authors should either delete or replace the reference.

3) Line 228: Fig. 3e should be Fig. 4e.

4) Lines 307, 309, 312, 316, and 317: Supplemental Fig. 1 should be Supplementary Fig. 3.

5) Line 428: A reference(s) should be provided for the role of ILC2 in asthma.

6) The authors used Student t-test, a one-way or two-way ANOVA followed by Sidak or Tukey post-hoc tests for statistical analyses. The method should be provided in the figure legends for each panel of the figures.

7) Legends for Fig. 1 and Supplementary Figs. 2, 3 and 4: The number of mice used should be provided.

Reviewer #2:

Remarks to the Author:

The authors have done a satisfactory job in addressing the previous concerns of this reviewer.

Reviewer #3:

Remarks to the Author:

The authors have made considerable effort to answer all of the reviewers' comments, performing a number of additional expts and discussing interesting points in detail. I have no further queries.

REVIEWERS' COMMENTS:

Reviewer #1 (Remarks to the Author):

Shafiei-Jahani et al. performed additional experiments and revised their paper, which was significantly improved. There are, however, several minor points that need to be addressed as listed below.

We thank the reviewer for their positive feedback about our revised manuscript. We have now addressed the minor points as kindly suggested by the reviewer below.

1) Lines 28 and 58: ILC2 are “group” 2 innate lymphoid cells but not “type” 2 innate lymphoid cells (PMID: 23348417, 30142344).

We changed “type 2” to “group 2” innate lymphoid cells in the manuscript.

2) Line 189: Ref. 29 is not related to insulin-resistance but to asthma. The authors should either delete or replace the reference.

We replaced the mentioned reference.

3) Line 228: Fig. 3e should be Fig. 4e.

We have now corrected the typo in the manuscript.

4) Lines 307, 309, 312, 316, and 317: Supplemental Fig. 1 should be Supplementary Fig. 3.

We have corrected the manuscript accordingly.

5) Line 428: A reference(s) should be provided for the role of ILC2 in asthma.

We added two references for the role of pulmonary ILC2s in asthma.

6) The authors used Student t-test, a one-way or two-way ANOVA followed by Sidak or Tukey post-hoc tests for statistical analyses. The method should be provided in the figure legends for each panel of the figures.

We have added the statistical methods to the figure legends.

7) Legends for Fig. 1 and Supplementary Figs. 2, 3 and 4: The number of mice used should be provided.

We have provided the number of mice used in the figure legends.

Reviewer #2 (Remarks to the Author):

The authors have done a satisfactory job in addressing the previous concerns of this reviewer.

We thank the reviewer for their positive feedback.

Reviewer #3 (Remarks to the Author):

The authors have made considerable effort to answer all of the reviewers' comments, performing a number of additional expts and discussing interesting points in detail. I have no further queries.

We appreciate the reviewer's positive feedback.